# LLM-Lasso: A Framework for Domain-Informed Feature Selection and Regularization

## Abstract

We introduce LLM-Lasso, a framework that leverages large language models (LLMs) to guide feature selection in Lasso $\ell_1$ regression. LLM-Lasso incorporates domain-specific knowledge extracted from natural language, optionally enhanced through a retrieval-augmented generation (RAG) pipeline, as penalty factors in the Lasso's $\ell_1$ regularization term. This integrates data-driven modeling with metadata from natural language. This is, to our knowledge, the first embedded LLM-driven feature selector. By design, LLM-Lasso addresses some key robustness challenges of LLM-driven feature selection: the risk of LLM hallucinations or low-quality responses. An internal cross-validation step is crucial to LLM-Lasso's performance, determining how heavily the prediction pipeline relies on the LLM's outputs. In the case of poor LLM generation quality, the cross-validation procedure allows LLM-Lasso to recover the behavior of the plain Lasso. In some biomedical case studies, LLM-Lasso outperforms standard Lasso and existing feature selection baselines.

## 1 Introduction

In statistical learning, feature selection enables models to focus on the most relevant predictors while reducing complexity and improving interpretability (Guyon et al., 2007; Chandrashekar & Sahin, 2014; Li & Liu, 2015). In particular, Lasso regression, which automatically selects a suitable linear model with a sparse set of coefficients, is interpretable and computationally efficient. Selection is performed by solving a straightforward convex optimization problem that promotes sparsity by penalizing the $\ell_1$ norm of the regression coefficients (Tibshirani, 1996; Bühlmann & Van De Geer, 2011; Hastie et al., 2015).

Lasso, as a traditional supervised learning model, develops a model soley based on the training data, It can be useful consider task-specific expert knowledge to inform the feature selection task. To do so, we take inspiration from Adaptive Lasso (Zou, 2006), which adds feature-specific weights to the $\ell_1$ regularization term of Lasso regression. Whereas adaptive Lasso commonly selects penalty factors based on weights from linear or ridge regression, LLM-Lasso instead aims to select penalty factors via domain knowledge. Prior works such as (Bergersen et al., 2011; Urda et al., 2018) assign penalties via external datasets or citation-based weighting to compute penalty factors. We take a similar but potentially more comprehensive approach: we leverage large language models (LLM) to generate penalty factors for all features.

The development of LLMs trained on large-scale unstructured text can be applied to augment traditional feature selection techniques. Transformer-based pre-trained LLMs, such as GPT-4 (OpenAI, 2023b) and LLaMA-2 (Touvron et al., 2023) have demonstrated impressive abilities in encoding domain knowledge and contextual relationships, generalizing to a wide range of unseen tasks (Vaswani et al., 2017; Brown et al., 2020; Radford et al., 2019; Manikandan et al., 2023), including: challenging reasoning tasks (Wei et al., 2022; Lewkowycz et al., 2022; Suzgun et al., 2023), domain-specific prediction tasks (Petroni et al., 2019; Dinh et al., 2022; Chen & Zou, 2024; Theodoris et al., 2023; Cui et al., 2024), and feature selection (Choi et al., 2022; Jeong et al., 2024; Li et al., 2024; Liu et al., 2024; Han et al., 2024). For instance, (Dinh et al., 2022) leverages LLM knowledge directly for regression problems by fine-tuning an LLM with training data, feature names, and task descriptions, showing comparable performance to traditional methods in low dimensions.

Efforts to incorporate LLMs into feature selectors have yielded filter and wrapper methods that perform competitively in low dimensions. (Choi et al., 2022) introduces the LMPriors framework, which selects features by analyzing log-probability differences when generating "Y" (Yes) or "N" tokens, with a context including task descriptions, feature names, and few-shot examples. For proprietary LLMs where internal token probabilities are inaccessible, (Jeong et al., 2024) proposes three prompting strategies that rely only on textual information. (Liu et al., 2024) provides a feature selection pipeline, starting with features selected by classical methods and using several epochs of LLM queries, informed by cross-validation accuracies for selected feature sets, to refine the features. As this is inherently wrapper-like, i.e., it doesn't see the metadata and data together, multiple iterations of LLM queries are required.

These LLM-guided feature selectors show promise for LLMs to encode task-specific knowledge. Current approaches, however, have yet to produce LLM-informed embedded feature selectors—methods that integrate feature metadata directly into regression models—and instead rely on multiple rounds of prompting or have LLMs directly select features. In particular, methods that select features based on LLM outputs alone are vulnerable to hallucinations, i.e., fabricated or inaccurate information, a well-documented weakness of LLMs (Huang et al., 2024; Yao et al., 2024). Such vulnerabilities raise concerns about reliability, especially in high-stakes domains like biomedicine, where data may be noisy or incomplete and precision is critical.

LLMs have also shown promise for learning directly from tabular data (e.g., (Hollmann et al., 2025)). Building on this capability, recent work has integrated tabular LLMs into statistical learning pipelines, for example by combining LLM predictions with decision trees (Jayawardhana et al., 2025) or by using LLMs as weak learners in boosting frameworks (Wang et al., 2025). While these approaches achieve strong empirical performance, they can be computationally expensive for large or high-dimensional datasets, since prediction requires running an LLM for each sample. In contrast, we incorporate LLM-informed feature regularization and leverage the efficiency of the Lasso to enable scalable end-to-end prediction.

**Main Contributions**    In this work, we introduce **LLM-Lasso**, a simple pipeline that incorporates knowledge encoded in textual feature metadata into supervised learning via adaptive Lasso regularization. Specifically, we use an LLM to generate feature-specific penalty factors, which are integrated directly into the Lasso objective. To guard against noisy or low-quality LLM outputs, we employ a cross-validated transformation of these penalties, allowing the method to adaptively control the influence of LLM-derived information.

We evaluate the performance of LLM-Lasso through a variety of experiments, focusing on high-dimensional oncology tasks where the number of features exceeds those in prior LLM-based feature selection studies by at least an order of magnitude. Our main contributions are as follows.

1. *We use contextual knowledge from LLMs to inform Lasso*, providing a scalable framework to directly integrate LLMs into traditional supervised learning. This framework can improve Lasso's accuracy, especially in a high-dimensional, low-data setting. We demonstrate this via large-scale experiments in the biomedical domain, where expert knowledge can be particularly important.
2. *We encourage robustness by using the data to cross-validate LLM decisions*, using $k$-fold cross-validation to choose from a family of transformations on the LLM-generated penalty factors. We construct the transformation family such that LLM-Lasso can never perform worse than Lasso on the cross-validation data. LLM-based filter feature selectors lack even this basic guarantee. Via adversarial examples in Section 4.4, we show that this can result in Lasso-like performance on the test data if LLM penalties are uninformative.

The paper is structured as follows: Section 2 reviews some preliminaries. Section 3 details our methodology. Section 4 evaluates LLM-Lasso on selected datasets and LLMs. Finally, Section 5 summarizes our findings.

## 2    Preliminaries

In this section, we provide some background on supervised learning, feature selection, and language modeling to contextualize LLM-Lasso.

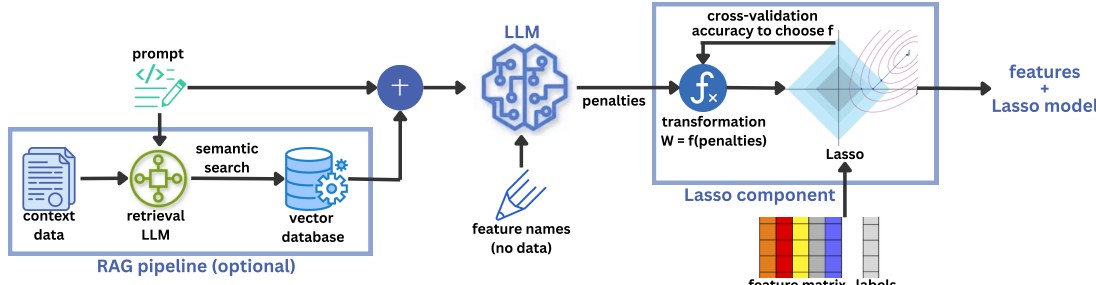

Figure 1: Full pipeline of collecting and using $\ell_1$-norm penalty factors in LLM-Lasso.

## 2.1 Supervised Data-Driven Learning

We consider a generic data-driven supervised learning procedure, with dataset $\mathcal{D}$ consisting of $n$ data points $x_i \in \mathbb{R}^p$ and targets $y_i \in \mathcal{Y}$. We consider univariate targets $\mathcal{Y} \subseteq \mathbb{R}$ (the regression setting) and $\mathcal{Y} \subset \mathbb{Z}$, $|\mathcal{Y}| < \infty$ (the classification setting).

Our goal is to represent the relationship between $x_i$ and $y_i$ by model $f_\theta : \mathcal{X} \to \mathcal{Y}$, where $\theta \in \Theta$ are the model parameters. $\theta$ is chosen to minimize learning objective

$$\hat{\theta} := g(\mathcal{D}) := \arg\min_\theta \mathcal{L}(\theta, \mathcal{D}) := \arg\min_\theta \sum_{i=1}^{n} \mathcal{L}(\theta, (x_i, y_i)), \tag{1}$$

where $\mathcal{L}$ is a loss function quantifying the error between predictions and true outcomes (e.g., cross-entropy loss or mean squared error with regularization).

**Feature Selection.** Feature selection aims to improve predictive performance while minimizing redundancy. Techniques generally fall into four categories: (i) filter methods, which rank features based on statistical properties, e.g., Fisher score (Duda et al., 2001; Song et al., 2012); (ii) wrapper methods, which evaluate model performance on different feature subsets (Kohavi & John, 1997); (iii) embedded methods, which integrate feature selection into the learning process(Tibshirani, 1996; Zou, 2006; Lemhadri et al., 2021); and (iv) hybrid methods, which combine elements of (i)-(iii) (Singh & Singh, 2021; Li & Ren, 2022). This paper focuses on embedded methods via Lasso, benchmarking against methods from (i)-(iii).

Related is the idea of model-specific feature relevance, which we discuss in Appendix A.

**Task-Specific Data-Driven Learning** In many settings, particularly high-dimensional low-data regimes, it is valuable to augment data-driven learning with *metadata*, or relevant domain expertise. This helps create accurate, reliable, and interpretable models, especially when pure data-driven models tend to overfit.

In this case, the estimator $g(\mathcal{D})$ from equation 1 becomes $\hat{\theta} = g(\mathcal{D}, \mathcal{D}_{\text{meta}})$, where $\mathcal{D}_{\text{meta}}$ is some form of metadata about the task. There are multiple approaches to this. For instance, LMPriors (Choi et al., 2022) and LLM-Select (Jeong et al., 2024) first specify which features to retain based on LLM-extracted feature importance, and then optimize over models restricted to those selected features. This approach selects important features solely based on metadata without seeing the dataset itself. In contrast, we directly embed task-specific knowledge into the optimization landscape by incorporating metadata-derived feature importance into the loss function's regularization term, as described in Section 3.

## 2.2 Language Modeling

Language modeling aims to approximate the true distribution of natural language. Modern large language models, trained on diverse datasets (Gao et al., 2020), exhibit strong generalization across domains, acquire contextual knowledge, and perform zero-shot learning—solving new tasks using only task descriptions—or few-shot learning by leveraging a small number of in-context examples (Brown et al., 2020).

**Retrieval-Augmented Generation (RAG).** Retrieval-Augmented Generation (RAG) enhances the performance of generative language models by integrating a document retrieval process (Lewis et al., 2020). The RAG framework comprises two main components: *retrieval*, which extracts relevant information from external knowledge sources, and *generation*, where an LLM generates responses using the prompt combined with the retrieved context. Documents are indexed through, e.g., relational, graph, or vector databases (Khattab & Zaharia, 2020; Douze et al., 2024; Peng et al., 2024), enabling efficient retrieval via semantic similarity search to match the prompt with relevant documents in the knowledge base. RAG has gained traction due to its empirical ability to reduce hallucinations and boost reliability (Huang et al., 2023; Zhang et al., 2023).

## 3  Methodology

We introduce the LLM-Lasso framework, consisting of two main components: (i) integration of LLM-informed penalties into Lasso; and (ii) LLM generation of penalty factors via prompt engineering and RAG. While the RAG component of (ii) is optional, it can substantially improve accuracy for certain tasks.

Figure 1 provides an overview of our framework.

### 3.1  The LLM-Lasso

We focus on the supervised learning framework introduced earlier in Section 2.1 with input feature $X \in \mathbb{R}^{n \times p}$ and response $Y \in \mathbb{R}^n$. The Lasso is a shrinkage method that places an $\ell_1$ penalty on the coefficients, causing some coefficients to be exactly zero. The loss function of the Lasso (Tibshirani, 1996) is:

$$\frac{1}{2}\sum\nolimits_{i=1}^{n}(y_i - \beta_0 - x_i^\top \beta)^2 + \lambda \sum\nolimits_{j=1}^{p}|\beta_j|. \tag{2}$$

To incorporate prior knowledge of the relationship between $X$ and $Y$ into the learning of a prediction model $g : (\mathcal{X} \times \mathcal{Y})^n \rightarrow \Theta$, one can enhance the Lasso by assigning penalty factors to each coefficient in the $\ell_1$ penalty (Zou, 2006). The loss function of the Lasso with penalty factors, or the adaptive Lasso, is:

$$\frac{1}{2}\sum\nolimits_{i=1}^{n}(y_i - \beta_0 - x_i^\top \beta)^2 + \lambda \sum\nolimits_{j=1}^{p}w_j|\beta_j|. \tag{3}$$

In LLM-Lasso, these penalty factors are generated by an LLM, with a data-driven cross-validation procedure to tune the adaptive Lasso's reliance on LLM knowledge.

#### 3.1.1  The Cross-Validation Procedure

Given the LLM-generated penalty factors $V \in \mathbb{R}^p$, we compute the final LLM-Lasso penalties $W^* \in \mathbb{R}^p$ by performing a transformation $\tau^* : \mathbb{R}^p \rightarrow \mathbb{R}^p$ on $V$. $\tau^*$ is chosen, using cross-validation, from a finite family of transformations, $\mathcal{T}$.

Via $k$-fold cross validation, we obtain $k$ different training and validation sets of the data. For split $i$, let $\beta_{i,W}^*$ be the optimal Lasso coefficients given the training data from the $i^{\text{th}}$ split and penalties $W = \tau(V)$. Let $X_{\text{val}}^{(i)}$ and $y_{\text{val}}^{(i)}$ refer to the validation data and labels for that split. Denoting the cross-validation loss function by $\mathcal{L}_{\text{CV}}$, the final LLM penalties are

$$W^* = \tau^*(V), \quad \text{where} \quad \tau^* = \arg\min_{\tau \in \mathcal{T}} \sum\nolimits_{i=1}^{k} \mathcal{L}_{\text{CV}}\left(X_{\text{val}}^{(i)} \beta_{i,\tau(V)}^*, y_{\text{val}}^{(i)}\right).$$

We define $\mathcal{T}$ to have a range of transformations representing different degrees of reliance on LLM-generated penalties. If the LLM penalties are high-quality, the cross-validation procedure chooses $\tau^*$ to accentuate the LLM penalties, i.e., increase the relative distance between low penalties and high penalties. By constructing $\mathcal{T}$ to include the null transformation $\tau_0(W) = 1$, LLM-Lasso can never do worse than plain Lasso in cross-validation loss. Though this is a basic, straightforward guarantee (and does not ensure generalization to test set performance), it provides a clear improvement in robustness over other LLM-guided feature selection methods, e.g., (Choi et al., 2022; Jeong et al., 2024).

### 3.1.2 Lasso and Cross-Validation Methodology

LLM-Lasso uses the adaptive Lasso implementation from the `adelie` (Yang & Hastie, 2024) `Python` library, which supports $k$-fold cross validation. We sweep the regularization strength, $\lambda$ with a $\lambda$ path size of 100 and default $\lambda_{\min}/\lambda_{\max}$ ratio of 0.001 for the large-scale experiments and 0.01 for the small-scale.

To choose the final $\lambda$ values, we use either misclassification rate or negative AUROC as the cross-validation metric (mean squared error for regression problems), depending on which metric we are plotting. If multiple $\lambda$ values result in the same number of features, we choose the one with the lowest cross-validation metric.

The loss function, $\mathcal{L}_{\mathrm{CV}}$, for choosing a transformation on the penalties, is the area under the "misclassification rate vs. number of features" plot, minus that of the Lasso. For experiments displaying the AUROC, we use the negative area under the "AUROC vs. number of features" plot instead.

**Downstream $\ell_2$ Heuristic** We empirically find that, for models with a small number of features (until $\approx$ 5-10) the linear models produced by Lasso perform noticeably worse than $\ell_2$-regularized regression (which we use as a downstream predictor for feature selection baselines). As a heuristic to counteract this, we run downstream $\ell_2$-penalized logistic regression on the LLM-Lasso- and Lasso-selected features for up to 5 features in the small-scale experiments and up to 10 features in the large-scale experiments.

Further methodological details can be found in the Results section (Section 4.1.2). Appendix E.1 contains a brief study of this heuristic and its interaction with the cross-validation component of LLM-Lasso.

### 3.2 Choosing a Transformation Family

We consider two broad families of transformations, and compare the two via a simulation experiment.

**Inverse Importance Family** [1] We define this transformation family as

$$\mathcal{T} = \left\{ \tau \,:\, \tau(V)_j = v_j^\eta, \eta \in \{0, 1, \ldots, \eta_{\max}\} \right\}.$$

For $\eta = 0$, the resulting penalties are the same as plain Lasso, indicating no reliance on the LLM output. Likewise, a high value of $\eta$ indicates heavy reliance on the LLM scores.

**Rectified Linear Unit (ReLU) Transformation Family** Let $V_{(j)}$ be the $j^{\mathrm{th}}$-largest penalty, and $\tau(V)_{(j)}$ be the corresponding index of the transformation $\tau(V)$. We set the largest penalty factor (the penalty factor of the least important feature) to a fixed value, $W_{(p)} > 1$. Then,

$$\tau(V)_{(j)} = \frac{(j - (1-\gamma)p)_+}{\gamma p} \cdot (W_{(p)} - 1),$$

where $\gamma \in (0, 1)$ is the ReLU threshold. The $(1-\gamma)p$ most important features receive a penalty of 0, and the rest receive a positive penalty that relates linearly to their LLM-derived feature ranking. The ReLU transformation family includes varying values of $\gamma$.

### 3.2.1 Transformation Family Simulation Results

We choose between these two transformation families via simulations on datasets outlined in Table 1. Based on the simulations, we use the inverse importance penalty factors for further experiments (Section 4).

We split each dataset into an importance score-generating set and a cross-validation set. Hypothetical importance scores are generated by running a Lasso regression on the score-generating set and setting importance scores $\mathcal{I}_j$ as the absolute value of the Lasso coefficient for the corresponding feature. $\mathcal{I}_j$ are scaled to be in the range $(0.1, 1)$, and reciprocated to produce penalty factors.

We perform prediction on the cross-validation set using Lasso, as well as LLM-Lasso with penalty transformations chosen from: (i) the inverse importance transformation family with $\eta \in (0, 1, \ldots, 10)$, and (ii) the

---

[1]Named as such based on the viewpoint of LLM-generated penalties as $v_j = \mathcal{I}_j^{-1}$, where $\mathcal{I}_j$ is the importance of feature $j$.

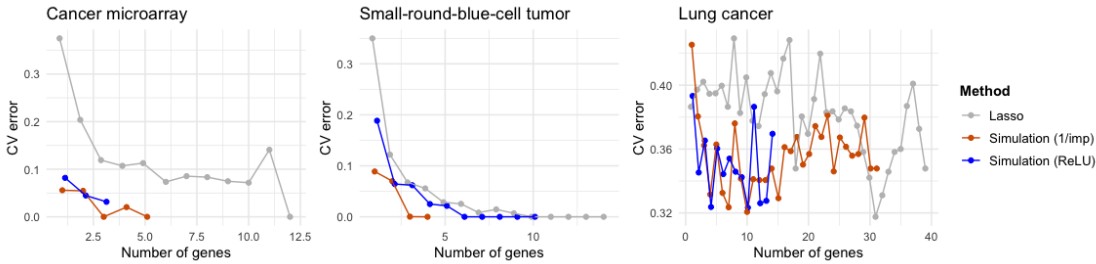

Figure 2: Test error in simulations

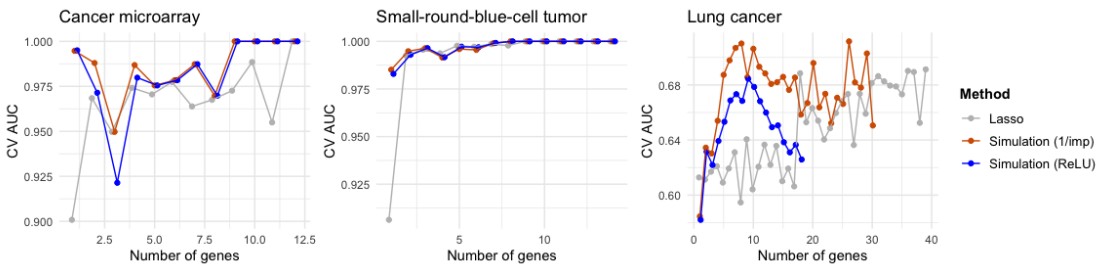

Figure 3: AUROC in simulations

| Dataset | Classification Task | n | p |
|---|---|---|---|
| Cancer microarray (Ramaswamy et al., 2001) | Rhabdomyosarcoma vs. Other cancers | 52 | 1000 |
| Small-round-blue-cell tumor (Khan et al., 2001) | Lymphoma vs. Leukemia | 83 | 1000 |
| Lung cancer (Spira et al., 2007) | Tumor vs. Healthy tissue | 187 | 1000 |

Table 1: Summary of simulation datasets for penalty factor form. We construct 1000-dimensional datasets from the original (higher-dimensional) tasks by selecting features with the highest variance.

ReLU transformation family with $\gamma \in (0.1, 0.2, \ldots, 0.9)$. For the ReLU family, we set the maximum penalty $W_{(p)}$ such that the least important feature receives a coefficient of 0 for all values of the Lasso regularization parameter, $\lambda$. We choose transformations via 5-fold cross-validation.

Figure 2 plots the cross-validation test error (using test error as the cross-validation metric), averaged over 10 different data splits (random seeds). Figure 3 shows the AUROC, with AUROC as the cross-validation metric. These simulations show an advantage of the inverse importance penalty factors over ReLU penalty factors and the Lasso. A possible explanation is that ReLU penalties only use the order of the importance scores as opposed to their relative magnitudes. Thus, we use the inverse importance penalty factors to compare the LLM-Lasso to baseline models in further experiments. Further exploration of transformation families is deferred to future work.

### 3.3 Task-Specific Language Modeling

#### 3.3.1 Prompt Engineering

Prompt engineering is an efficient and effective approach for adapting pretrained LLMs to tackle new tasks not encountered during training (Radford et al., 2019; Liu et al., 2023). In our experiments, we employ a zero-shot approach (i.e., including only background information and task instructions in our prompt), incorporating chain-of-thought (CoT) prompting (Wei et al., 2022), a technique shown to significantly enhance performance on complex reasoning tasks.

Our main focus is the *user query*, a task description prompt with descriptions of features and target variables. For RAG-augmented LLM-Lasso, the *retriever* is also relevant. The top $k$ documents are retrieved via semantic similarity to a retrieval prompt (which includes feature names and a brief task description) and appended to the user prompt.

The user prompt follows the structure in Figure 4, where it is composed of a background description of the dataset, the assigned task, and formatting instructions. We refer the readers to Appendix B for a more detailed description of the specific prompts used in our experiments.

### 3.3.2 Knowledge Base Embedding via RAG

We use a standard RAG pipeline to optionally embed a task-specific knowledge base for our oncology prediction tasks. RAG provides relevant contextual information by retrieving documents from a database—crucial given LLMs' limitations with long contexts. Below, we outline our RAG pipeline.

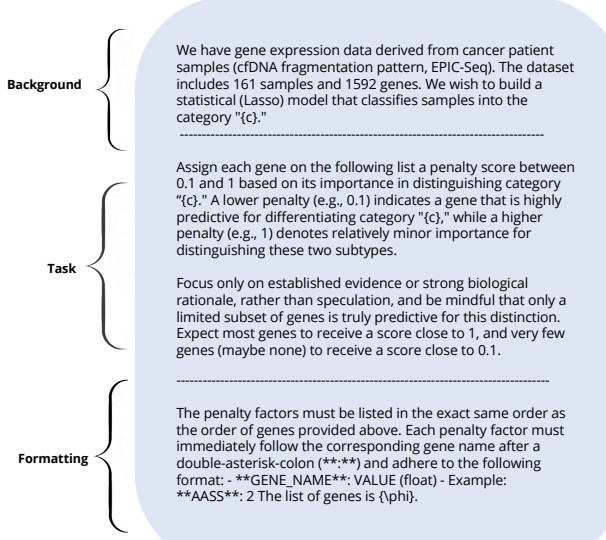

Figure 4: An example user query prompt.

**Preprocessing** Given a knowledge base of $N$ text documents, $\{D_i\}_{i=1}^N$, we obtain their $d$-dimensional semantic embeddings $\{d_i\}_{i=1}^N = \{E(D_i)\}_{i=1}^N$ via an embedding function $E : \text{Text} \to \mathbb{R}^d$. Here, we use the `OpenAI` embeddings off-the-shelf (OpenAI, 2024). Upon obtaining the semantic embedding vectors we apply the the Hierarchical Navigable Small World (HNSW) algorithm (Malkov & Yashunin, 2018), implemented in `chromadb`, to enable sublinear complexity for semantic similarity search.

**Retrieval** At retrieval time, given a query vector $q \in \mathbb{R}^d$, the semantic similarity between $q$ and the stored embeddings $\{d_i\}_{i=1}^N$ is computed via the cosine similarity function. The top $k$ documents with the highest similarity scores are appended to the user prompt.

Throughout the paper, `LLM-Lasso (Plain)` refers to a pipeline without RAG, while `LLM-Lasso (RAG)` incorporates RAG. The performance of RAG in our framework highly depends on the retrieval prompt and the relevance of the retrieved documents. Though a more detailed study of RAG is reserved for future work, our RAG system empirically boosts LLM-Lasso accuracy in the large-scale biomedical examples.

## 4 Experiments

In this section, we evaluate LLM-Lasso through a series of experiments. These include small-scale tests ($\sim 20$ features) and large-scale biomedical experiments ($> 1000$ features). We demonstrate the following:

1. *In the right setting, metadata improves Lasso performance* (Section 4.2): Large-scale biomedical experiments highlight a domain where LLM-Lasso penalty factors can outperform an array of baselines. Notably, in this setting, LLM-Score performance is degraded compared to small-scale experiments.
2. *LLM-Lasso, though not always the best model, is competitive for a variety of tasks* (Section 4.3): In a variety of small-scale experiments, LLM-Lasso is competitive against our array of baselines.
3. *LLM-Lasso can be more robust than comparable LLM-driven methods* (Section 4.4): In adversarial simulations with corrupted feature names, where LLM-Score does noticeably worse than random feature selection, LLM-Lasso performance approximates that of the plain Lasso.

Refer to Appendix E for additional experimental results: a brief study of the downstream $\ell_2$-regularized regression heuristic, AUROC results for the large-scale experiments, an ablation of model temperature,

document recall evaluation for the RAG system, and a study of the features selected by LLM-Lasso vs. Lasso for a large-scale oncology dataset.

## 4.1 Experimental Details

### 4.1.1 Large Language Models Used

For the experiment, we sample a combination of closed-source and open-source LLMs, as described in Table 2. We use all GPT models via `OpenAI` API calling and all open-source models via `OpenRouter` API calling via cloud-based inference. We implement RAG using the `langchain-community` (LangChain Community, 2024) library and a self-query retriver as our base method for query construction via `Chroma` vectorstore.

Table 2: LLMs used in LLM-Lasso experiments.

| Model | o1 | GPT-4o | GPT-3.5 | DeepSeek-R1 |
|---|---|---|---|---|
| **Citation** | (OpenAI, 2024) | (OpenAI, 2023b) | (OpenAI, 2023a) | (DeepSeek AI, 2025) |
| **Parameters** | $-^2$ | $-$ | $-$ | 671B |
| **Cut-off Date** | 2023.10 | 2023.10 | 2021.09 | 2024.07 |

| Model | LLaMA-3.1 | LLaMA-3 | Qwen | |
|---|---|---|---|---|
| **Citation** | (Meta AI, 2025b) | (Meta AI, 2025a) | (Alibaba DAMO Academy, 2025) | |
| **Parameters** | 405B | 8B | 72B | |
| **Cut-off Date** | 2023.12 | 2023.12 | 2023.09 | |

For the large-scale experiments, o1 is used for DLBCL vs. FL and LUAD vs. LUSC, and GPT-4o is used for DLBCL vs. MCL. The small-scale experiments all use GPT-4o. On any given dataset, the same model is used for LLM-Lasso, LLM-Score, and LMPriors (where applicable).

### 4.1.2 LLM-Lasso Evaluation Methodology

To test the prediction performance of the LLM-Lasso, the data is centered and equally split into the training set and the test set. For classification problems, a stratified split is used.

On the training set, we perform $k$-fold cross-validation across the hyperparameter $\eta \in (0, 1, 2, \ldots, \eta_{\max})$ for penalty factors of the inverse importance form, $V^\eta$. We use $\eta_{\max} = 4$ by default. The o1 experiments have a more intensive cross-validation step (see **Additional Heuristics** below), so we use $\eta_{\max} = 1$ instead to reduce compute time. For cross-validation, we use $k = 5$ for the large-scale experiments and $k = 10$ for the small-scale experiments, as the large-scale experiments have relatively few datapoints (they are called *large-scale* due to their high dimensionality, but are underdetermined).

Cross-validation and the corresponding metrics is as described earlier in Section 3.1.2. As a robustness check, we repeat the process across 10 random splits and plot the mean.

**Building a Knowledge-Base for RAG** We use OMIM (Online Mendelian Inheritance in Man), an open-source database of human genes and associated diseases, to construct our RAG knowledge base. Gene symbols, clinical synopses, and genetic-phenotypic relationships are extracted via the OMIM API and stored in `JSON` format. This `JSON` data is then chunked with a recursive text splitter (implemented through `langchain-community`) and indexed into a `Chroma` vector store.

**Handling Token Limits in LLMs** Input and output token limits in pretrained LLMs pose significant challenges, especially when generating scores for large sets of predictors, as they restrict the information processed or returned in a single interaction. Ensuring that an LLM has sufficient output tokens is critical for its performance. For instance, (Wei et al., 2022) showed that step-by-step reasoning improves effectiveness, and we observe that LLMs struggle under tight token limits or when limits are exceeded.

To handle large feature sets (e.g., $p \approx 1000$), we perform batch querying, asking the LLM to produce a subset of scores for each query. For the small-scale experiments, all scores can be collected in one query. To

balance context length with number of queries, we heuristically set the batch size to be $\approx \sqrt{p}$ (30 for the lung cancer dataset, and 40 for the lymphoma datasets).

To encourage consistent scores, we constrain the penalty factors to be in a pre-determined range, which we encode in our prompt. We set this range to be between 0.1 and 1 for the large-scale datasets and between 2 and 5 for the small-scale datasets.

**Additional Heuristics** For experiments with `o1`, where we are not permitted to set the LLM generation temperature to $T = 0$, we collect scores across 3 trials and use cross-validation to select the best set of scores. For the empirically-calibrated prompt, we find that the range of the scores is not conducive to optimal Lasso performance. Accordingly, we add a constant factor of 2 to each penalty factor before applying the inverse importance transformation (placing the penalties in the $[2.1, 3]$ range).

### 4.1.3 Baselines

We compare LLM-Lasso against baselines from both LLM-based feature selectors and traditional data-driven feature selection methods: (1) LLM-Score (Jeong et al., 2024), (2) LMPriors (Choi et al., 2022) [3], (3) Filtering by Mutual Information (MI) (Lewis, 1992), (4) Recursive Feature Elimination (RFE) (Guyon et al., 2002), (5) Minimum Redundancy Maximum Relevance selection (MRMR) (Ding & Peng, 2005), (6) Lasso (Tibshirani, 1996), (7) XGBoost (Chen & Guestrin, 2016), (8) Sequential Attention (Yasuda et al., 2023), (9) LassoNet (Lemhadri et al., 2021), (10) Random feature selection.

For the large-scale experiments, we include two ablation variants alongside the baselines. The first, Adaptive Lasso (Zou, 2006), assigns Lasso weights based on the (reciprocal of the) final feature weights from an $\ell_2$-regularized logistic regression model. The transformation of these scores is selected using the same cross-validation procedure employed for the LLM-Lasso weights. The second variant, which we call "LLM-Score Weights", omits the cross-validation framework and directly applies the inverse of LLM-Score importance scores as penalty weights in the weighted Lasso.

For feature selection baselines other than Lasso, we follow the procedures outlined in (Jeong et al., 2024) to ensure a fair comparison: their performance is evaluated by measuring the test performance of a downstream $\ell_2$-penalized logistic regression model, with hyperparameters chosen via grid search and cross-validation.

## 4.2 Large-Scale Experiments

Gene expression levels support cancer diagnosis and prediction, while identifying predictive genes deepens our understanding of cancer and aids drug discovery. To showcase the applicability and strong performance of LLM-Lasso on high-dimensional, complex data, we evaluate it on cancer diagnosis and classification tasks using gene expression features. We focus on biomedical data to showcase a domain with high-dimensional, low-data learning problems where domain knowledge is both abundant and useful.

### 4.2.1 Datasets

To address concerns on LLM memorization and ensure transparency and reproducibility, we conduct experiments on both an unpublished lymphoma dataset and a publicly available lung cancer dataset.

**Lymphoma (Unpublished)** Follicular lymphoma (FL) is a relatively indolent form of lymphoma that usually does not require intervention, but it could occasionally transform into the more aggressive diffuse large B-cell lymphoma (DLBCL). Using an unpublished dataset of 1592 gene expression levels from 130 lymphoma samples, we use LLM-Lasso to classify tumor samples into DLBCL and FL. Though less clinically significant, we also perform the task of classifying 161 lymphoma samples into DLBCL and mantle cell lymphoma (MCL) using 1592 gene expression levels.

---

[3]We only consider the LMPriors baseline for the small-scale experiments, as it is cost-prohibitive to perform one API query per feature for high-dimensional problems.

**Lung Cancer (Public)** We additionally perform evaluations on an open-source lung cancer dataset. Data are obtained from The Cancer Genome Atlas Program (TCGA) (Weinstein et al., 2013), a publicly available database of human tumors. Our sample consists of bulk RNA sequencing data from 516 samples from patients with lung adenocarcinoma (LUAD) and 501 samples from patients with lung squamous cell carcinoma (LUSC). Only primary tumor samples are used. The data are normalized, variance-stabilized, and transformed using DESeq2 (Love et al., 2014). If more than one sequencing data is available for a patient, the average of the counts are taken. Genes with fewer than 10 counts are filtered out. We use the top 1000 most variable protein-coding genes in the downstream analyses.

### 4.2.2 Results

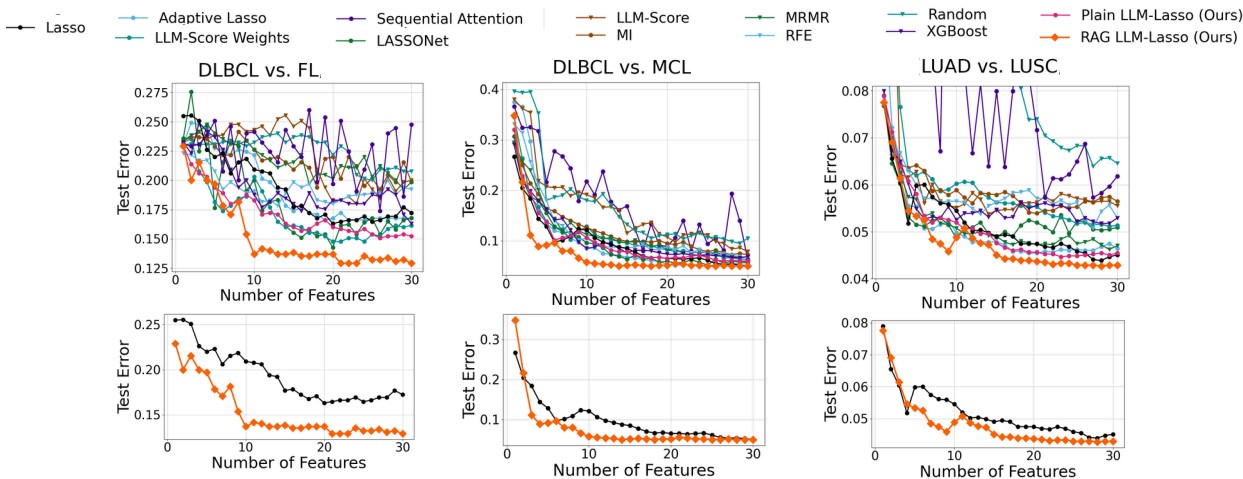

Figure 5: LLM-Lasso vs. Baselines on high-dimensional biomedical classification tasks. LLM-Lasso vs. Lasso is plotted separately to illustrate the gains of using LLM-generated features in this setting.

Results from large-scale experiments are shown in Figure 5. Notably, LLM-Lasso with RAG consistently achieves lower test error compared to all sampled baselines and across all three datasets. Figure 6 shows the test error "win ratio" between LLM-Lasso and Lasso, i.e., the ratio of points in the first 30 features where RAG LLM-Lasso is strictly better than Lasso, and vice versa.

The two ablations (Adaptive Lasso and LLM-Score Weights) highlight the utility of key design features: namely, the use of an LLM to produce penalty factors and the choice of a transformation via cross-validation. Both ablation baselines perform well in some cases, but neither outperforms LLM-Lasso across the board. The LLM-Score Weights baseline does well (on-par with no-RAG LLM-Lasso) for the DLBCL vs. FL dataset, but is significantly worse for the other two datasets. Adaptive LASSO has the opposite behavior (strong for DLBCL vs. MCL

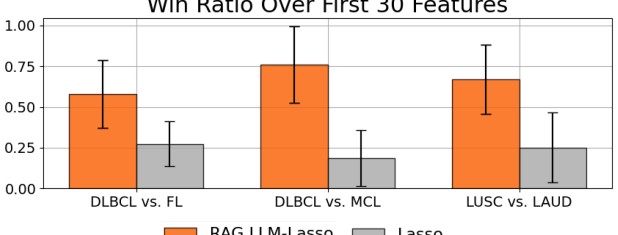

Figure 6: Win ratio between LLM-Lasso and Lasso over 10 random seeds, with standard deviation error bars.

and LUAD vs. LUSC but weaker for DLBCL vs. FL). These ablations show that the cross-validation procedure indeed helps robustness, as in two out of three of the large-scale experiments, simply adding LLM-Score-derived weights to weighted LASSO model performed worse than plain LASSO.

Overall, in the large-scale experiments, LLM-Lasso achieves strong performance, given that the LLM prior is informative enough. For such biomedical use cases, the curation of a RAG knowledge base greatly helps the informativeness of this prior.

### 4.3 Small-Scale Experiments

We also evaluate LLM-Lasso using small-scale, low-dimensional public datasets across various domains: three binary classification datasets (`Bank`, `Diabetes`, `Glioma`) and two regression datasets (`Wine Quality`, `Spotify 2024`). See Appendix D for more details on these datasets. Note that due to presentation clarity and our focus on the large-scale experiments, we choose a representative subset of the aforementioned baselines here. As shown in Figure 7, GPT-4o-based LLM-Lasso is consistently competitive, even though it is not always the best model. As these small-scale tasks lie in a low-dimensional, high-data regime, data-driven learning alone performs highly, with LLM-Lasso providing only marginal gains over plain Lasso. Indeed, LLM-Lasso can perform very well on tasks where metadata information is especially useful (e.g., the high-dimensional biomedical examples of Figure 5), but it is by no means the best model across all scenarios.

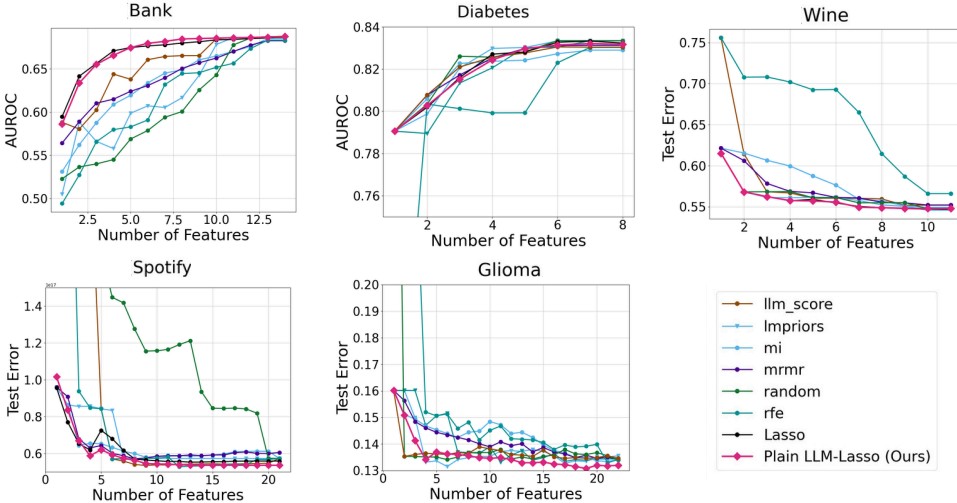

Figure 7: Small-scale experiments using GPT-4o. **Note**: In Glioma and Wine, the LLM-Lasso and LASSO lines fully overlap, due to cross-validation choosing to perform the null transformation on the LLM-generated penalties.

### 4.4 Adversarial Simulations

To showcase our robustness compared to pure LLM-based feature selection, e.g., (Jeong et al., 2024), we perform an adversarial data corruption simulation. As a base dataset, we use the DLBCL vs. MCL task from the Lymphoma dataset. Of the 1592 genes, we select the top 800 based on presence in documents retrieved from our OMIM RAG system (see Section 4.1.2). We replace those genes with random base64 strings, ensuring via OMIM that the strings are not real gene names (see Figure 8a).

To highlight the Lasso-like behavior of LLM-Lasso in this scenario, we *do not perform downstream $\ell_2$-regularized regression (Section 3.1.2) for Lasso or LLM-Lasso,* as the experiments in Appendix E.1 indicate that this heuristic interacts with the cross-validation in unexpected ways.

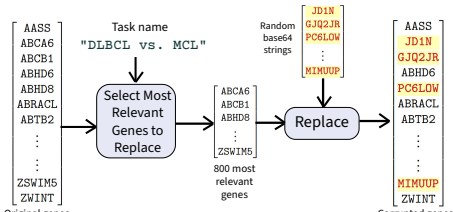

(a) Gene name corruption for adversarial simulations.

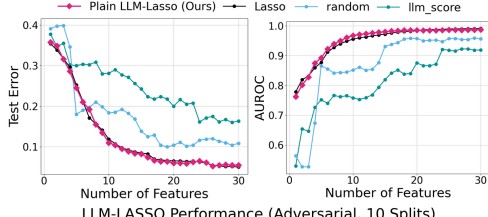

(b) Adversarial simulation results: DLBCL vs. MCL.

Figure 8: Adversarial simulation methodology diagram (left) and results (right).

We run LLM-Lasso and LLM-Score using GPT-4o, with both methods being given the corrupted gene name list. We also include a random feature selection baseline. The resulting misclassification error and AUROC are plotted in Figure 2. We observe that the LLM analysis of corrupted genes is heavily based on hallucinations, examples of which are in Figure 15 in the Appendix. Even so, the accuracy of LLM-Lasso remains comparable to Lasso, whereas LLM-Score performs noticeably worse than random feature selection. As the LLM output is un-insightful, we cannot expect LLM penalties to improve on Lasso; what we show is that, even in this adversarial case, performance is still on-par with the plain Lasso.

### 4.5 Model Ablations

Figure 9 performs LLM-Lasso with LLMs listed in Section 4.1.1: it displays the average misclassification error and AUROC of LLM-Lasso at 20 features for DLBCL vs. MCL, and the average mean squared error at 5 features for Spotify (over 10 splits). Lasso is plotted as a baseline, and all methods have standard deviation error bars. Larger, more powerful models generally perform better, especially with RAG. Some key exceptions are o1, which does worse than the smaller GPT-4o for DLBCL vs. MCL, and DeepSeek-R1, for which RAG degrades performance. We hypothesize that DeepSeek-R1 is harmed by the increased context from RAG, whereas other models may have a more nuanced ability to parse scientific texts.

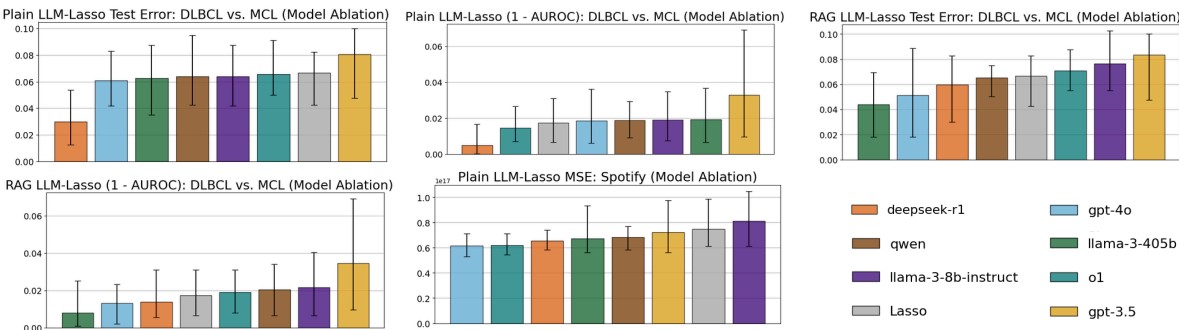

Figure 9: Model Ablations for DLBCL vs. MCL (large-scale) at 20 features, and Spotify (small-scale) at 5 features.

## 5 Discussion and Conclusion

LLM-Lasso is a simple, tunable model that incorporates domain-specific knowledge from LLMs. It achieves strong performance with a small number of features, with the potential to improve predictive accuracy by highlighting informative variables. LLM-Lasso safeguards against LLM hallucinations through cross-validation, which allows the model to modulate its reliance on LLM-derived penalty factors.

LLM-Lasso has several limitations that suggest directions for future work. First, LLM querying costs could be reduced. Second, our current focus on linear models limits expressiveness; extending the approach to nonlinear feature selectors, structured sparsity (e.g., group Lasso), or other sparsity-inducing penalties, could improve accuracy and adaptability. Potentially, in the structured sparsity case, LLMs could inform both group structure and feature importance. Third, key design choices, such as the transformation family, prompting strategy, and RAG database construction, warrant deeper study to better understand their impact on performance and generalization. Finally, theoretical analysis could enable stronger robustness guarantees.

## 6 Broader Impact

Though this work shows promising initial results on biomedical data, it is not ready for direct clinical use. Further validation of LLM-Lasso's robustness and performance is required prior to adoption. The reliance on LLMs may also perpetuate unwanted biases from the biomedical literature, with the usage of closed-source LLMs raising transparency and reproducibility concerns. Finally, the primary large-scale evaluation in this paper is on biomedical data; for practitioners in other domains, further evaluation is necessary.

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

## A  Model-Specific Feature Relevance

Feature importance can be extracted directly from statistical models. For instance, the magnitude of coefficients in Lasso (Tibshirani, 1996), ridge (Hoerl & Kennard, 1970), and elastic net (Zou & Hastie, 2005) regressions can be directly interpreted as feature importance, given that the features are standardized. Feature importance of tree-based methods, in general, is based on the improvements in accuracy brought by splits on a specific feature. For instance, XGBoost uses a *gain* metric to quantify the improvement in accuracy from a split (Chen & Guestrin, 2016), while random forests use mean decrease impurity, which measures the total reduction in impurity by all splits on a given feature (Breiman, 2002). Use of random permutations of features to evaluate feature importance (Breiman, 2001) is also popular.

## B  Prompt Engineering

Prompting is shown to be significant to the performance of LLMs. In this section, we detail design choices we made in designing the prompts for LLM-Lasso, and present the exact prompts used in the experiments.

### B.1  System Prompt

Throughout our large-scale experiments with the biomedical datasets, we set the system message to the generation LLM as "assistant," with instruction: "you are an expert assistant with access to gene and cancer knowledge." For the small-scale experiments, we do not set a custom system prompt.

### B.2  User Prompt

The general format of the user prompt follows Figure 4 in the main body. However, there are many ways one can format each of the three sections, that is, background description, a task description, and formatting rules. In the following, we go through each component in depth.

**Background Description.**  The background description includes two key components: (1) *dataset metadata*, which includes details on how the data is collected, number of samples, and number of features; and (2) *user intention*, or a description of our goal for data analysis. For instance, if we are conducting classification experiments using LLM-Lasso, the user intention is, e.g.: *"We wish to build a statstical (Lasso) model that classifies samples into category diffuse large B-cell lymphoma (DLBCL) and follicular lymphoma (FL)."*

**Task Description.**  The task section specifies the exact request made to the generation LLM. For LLM-Lasso, this involves a instructions of how to determine the penalty factors. Through experimentation, we found that a relatively direct and straightforward prompt works best. Specifically, we construct a prompt based on features of the LLM-generated penalty factors that empirically result in better LLM-Lasso performance. In addition, we consult advanced LLMs, such as o1 from `OpenAI` for advice on tuning this prompt in a way that would be most conducive for LLM performance.

> Assign each gene on the following list a penalty score between 0.1 and 1 based on its importance in distinguishing "{category}". A lower penalty (e.g., 0.1) indicates a gene that is highly predictive for differentiating "{category}", while a higher penalty (e.g., 1) denotes relatively minor importance for distinguishing these two subtypes.
>
> Focus only on established evidence or strong biological rationale, rather than speculation, and be mindful that only a limited subset of genes is truly predictive for this distinction. Expect most genes to receive a score close to 1, and very few genes (maybe none) to receive a score close to 0.1. If a gene is only generally relevant to {broader_topic} and does not specifically help distinguish "{category}", assign a penalty closer to 1.

This prompt is constructed to address several factors that can contribute to low-quality LLM scores, including reliance on speculation rather than known biological associations, over-assignment of low penalties, and assignment of scores based on broader relevance rather than the specific classification task. We note that the statement, "expect most genes to receive a score close to 1" is specific to high-dimensional cases where only a subset of features are predictive. For tasks, such as the small-scale datasets, where most features are

relevant, this description should be omitted. In addition, the range $[0.1, 1]$ is arbitrary and can be chosen by the user. In fact, we find that post-processing these scores to lie within a range farther away from 0 helps downstream Lasso performance, as noted in our experimentation methodology (Section 4.1.2).

**Task Description for Small-Scale Experiments.** As the above prompt is tuned for high-dimensional problems where only a small percentage of features are predictive, it is not relevant to the small-scale experiments. For the small-scale experiments, we instead use the following task description:

> Provide penalty factors for each of the features. These penalty factors should be integers between 2 and 5 (inclusive), where: 2 indicates a feature strongly associated with the target variable (i.e., it should be penalized the least by Lasso). 5 indicates a feature with minimal relevance to the target variable (i.e., it should be penalized the most by Lasso). Focus only on established evidence or strong rationale, rather than speculation. Focus on features that are immediately predictive for the task at hand, rather than those that are more generally relevant.

Again, the range $[2, 5]$ is arbitrary; there is no specific reason it was chosen here instead of $[0.1, 1]$. We also include a short description of each feature in the small-scale experiments, placed at the end of the prompt right before the full list of features.

**Output Format Instructions.** Our experiments revealed that selecting appropriate output format instructions is crucial not only for the accuracy of the score collection process but also for maintaining the quality of the scores produced. This is especially important for smaller models with fewer parameters (e.g., `llama-3-8b-instruct`), which often struggle to follow prompt instructions and understand the concepts and guidance provided.

In practice, we found that directly using text responses and providing LLMs with clear text formatting rules is more effective in regulating their behavior and ensuring a smooth score collection process compared to requesting responses in raw `JSON` format, as commonly used in `LangChain`'s pipeline. For all LLMs, we attach a format instruction to the end of every prompt, with slight modifications tailored to the specific task. Below is an example of a format instruction used for the task of outputting penalty factors for gene selection in cancer or lymphoma prediction.

> Formatting Rules:
> 1. Score Representation: Use a direct floating-point number (e.g., 0.5). Avoid scientific notation (e.g., 10**(-2) or 1e-2) and additional formatting.
>
> 2. Include All Genes: Assign a penalty factor for every gene in the input list, preserving the order of input.
>
> 3. Reasoning: After each penalty factor, add a concise reasoning about the gene's role in predicting {category}.
>
> 4. Consistency: Ensure uniform formatting. Example:
>
>     AASS: 0.15
>     Reasoning: This gene is highly expressed in cancer pathways and has been associated with {category}. Assigned a low penalty factor.
>     BRCA1: 1
>     Reasoning: BRCA1 is not significantly relevant for {category}. Assigned a high penalty factor.
> Do not include disclaimers about lacking full data; rely on general cancer genomics and pathway relevance.

As outlined in the formatting prompt, three strategies were found to be particularly effective:

1. *Highlighting common errors*: We include a list of frequent formatting mistakes made by LLMs, identified through trial and error. These include, for example, using scientific notation instead of floating-point numbers, which complicates the score collection algorithm, and applying inconsistent additional formatting to the scores.

2. *Providing examples*: Examples demonstrating the desired score and explanation format significantly improve the LLMs' understanding of the task. This is particularly important when querying for penalty factors instead of importance scores: smaller models like `llama-3-8b-instruct` often struggle with the counterintuitive nature of penalty scores, where lower values indicate higher significance and vice versa. Including examples of both low and high penalty scores helps address this challenge and ensures better compliance.

3. *Using a firm tone*: We employ strict language to enforce adherence. Commands such as "Do not say that it's not possible..." and "Responses not following these guidelines will be considered invalid" have proven effective in ensuring LLMs behave consistently and follow the guidance provided.

Figure 10 is an example of the full user prompt used in the study of classifying patients into DLBCL and FL, which employs `o1` to edit the text of the prompt.

*Context**: We have gene expression data derived from cancer patient samples (cfDNA fragmentation pattern, EPIC-Seq). The dataset includes 161 samples and 1592 genes. We wish to build a statistical (Lasso) model that classifies samples into the category "diffuse large B-cell lymphoma (DLBCL) and follicular lymphoma (FL)."

**Task**: Assign each gene on the following list a penalty score between 0.1 and 1 based on its importance in distinguishing "diffuse large B-cell lymphoma (DLBCL) and follicular lymphoma (FL)". A lower penalty (e.g., 0.1) indicates a gene that is highly predictive for differentiating "diffuse large B-cell lymphoma (DLBCL) and follicular lymphoma (FL)", while a higher penalty (e.g., 1) denotes relatively minor importance for distinguishing these two subtypes.

Focus only on established evidence or strong biological rationale, rather than speculation, and be mindful that only a limited subset of genes is truly predictive for this distinction. Expect most genes to receive a score close to 1, and very few genes (maybe none) to receive a score close to 0.1. If a gene is only generally relevant to lymphoma and does not specifically help distinguish 'diffuse large B-cell lymphoma (DLBCL) and follicular lymphoma (FL)", assign it a penalty closer to 1.

The penalty factors must be listed in the exact same order as the order of genes provided above. Each penalty factor must immediately follow the corresponding gene name after a double-asterisk-colon (**:**) and adhere to the following format:

- **GENE NAME**: VALUE (float) - Example: **AASS**: 2
**Instructions**:
1. You will receive a list of genes: {genes}.
2. For each gene, produce an floating point penalty factor from 0.1 to 1.
3. List the genes and their penalty factors in the exact same order they appear in the list.
  - Note that letter "I" is not number "1", so do not write "ARSI" as "ARS1".
  - For each gene, include ALL its letters: for instance, "BYSL" is NOT "BYS".

4. For each penalty factor, provide a brief statement of how you arrived at that factor or why the gene is more or less relevant to "diffuse large B-cell lymphoma (DLBCL) and follicular lymphoma (FL)."

Do not include disclaimers about lacking full data; rely on general cancer genomics and pathway relevance.

The list of genes is ["AASS", "ABCA6", "ABCB1", "ABHD6", ...].

Figure 10: Example of a full user prompt for experiment study DLBCL vs FL.

### B.2.1 Retrieval Prompt

The default pipeline in `Langchain` for retrieval query is to perform semantic similarity search on the user's original prompt to the generation LLM. This becomes problematic, however, when the main user prompt is large and overshadows the important information that sheds light on what documents should be retrieved. As an example, when passing in directly the full user prompt for retrieval in the oncology classification tasks, semantic similarity search retrieves information on the description of the dataset, for example, contexts regarding cfDNA fragmentation pattern and EPIC-Seq, instead of what we are actually curious about, that is, the relevance of certain gene, say AASS, with classifying lymphoma subtypes, say, diffuse large B-cell lymphoma (DLBCL) and follicular lymphoma (FL).

In order to pinpoint the retriever to the relevant retrieval documents, we use a customized retrieval prompt. For the oncology classification tasks, due to the high-dimensional nature of the dataset, we batch process the genes (see Section 4.1.2 for discussion) and use the following prompt that takes each gene $g_i \in \{g_1, ..., g_B\}$ in each batch of size $B$ and the target classification category $c$:

Retrieve information about gene {g}, category {c}, especially in the context of {g}'s relevance to {c}.

An example prompt using this format is as follows:

Retrieve information about gene AASS, category "transformed follicular lymphoma (tFL) and follicular lymphoma (FL)", especially in the context of AASS's relevance to "transformed follicular lymphoma (tFL) and follicular lymphoma (FL)".

For each pass of retrieval search with gene and lymphoma pair, we retrieve top $k$ relevant documents. After collecting the contexts for all the genes in the batch, we then filter for the unique documents and then append them to the full prompt in prompt component. In this process, we aim to strike a balance between overwhelming the generation LLM with long-context and potentially minimally informative documents and excessive cautious retrieval that does not inform the LLM by much.

## C   Implementation Details

### C.1   Computational Resources

**LLM Query Cost**   The LLM score-collection process requires a number OpenAI or OpenRouter queries equal to the number of batches of features, times the number of trials. The large-scale experiments perform score collection in batches of $\approx \sqrt{n}$ genes, requiring $\approx 40$ batches for each large-scale experiment. The experiments using the `o1` model are performed with 3 trials, and the experiments using `GPT-4o` are performed with one trial (as the temperature parameter of the LLM was set to 0).

For DLBCL vs. FL, where we used 3 trials and the more expensive o1 model, the API token cost for LLM-Lasso was \$11.99 without RAG and \$19.26 with RAG. The wall-clock time with 5 threads was 21 minutes without RAG and 42 minutes with RAG. For DLBCL vs. MCL, where we used only one trial and the cheaper GPT-4o model, the API token cost was \$0.73 without RAG and \$1.18 with RAG. The time cost (with the same level of parallelism) is 3 minutes without RAG and 9 minutes with RAG.

**LLM Querying and RAG Runtime**   Scores are collected with 5 threads performing API calls in parallel, though this is only an optimization to reduce experiment runtime. Model ablation studies only used one trial regardless of model type (with temperature set to 0 where allowed), due to the large volume of LLM queries needed otherwise.

Populating the OMIM vectorstore via `Chroma` takes around 5 minutes on a MacBook M1 10-chord CPU. For RAG-augmented LLM Lasso, querying the OMIM RAG database for the 1592-gene dataset takes about 10 minutes per experiment, resulting in 30 minutes for the large-scale experiments and 70 minutes for the model ablations.

**Lasso and Cross-Validation Runtime**   The Lasso component of LLM-Lasso is run on a laptop with 16 `AMD Ryzen 9 5900HX` processors and 16 GB of memory. Lasso and logistic regression models are run using 8 threads (implementation details of which are handled by `adelie` and `scikit-learn`, respectively). Each downstream experiment (LLM-Lasso, for a single dataset with 10 different random seeds) takes approximately 15 minutes for experiments with multiple trials of LLM prompting (where the cross-validation step is more involved) and 5 minutes otherwise.

### C.2   `R` implementation

Once the importance scores are obtained from the LLM, the LLM-Lasso can be implemented in `R` (R Core Team, 2024). One can pass the penalty factors, transformed into the form of choice, such as the ReLU-form or inverse importance and their powers ($\mathcal{I}^{-\eta}$), into the `cv.glmnet` function in the package `glmnet` (Friedman et al., 2010). The penalty factors can be passed into an argument called `penalty.factor`, which specifies the penalty factors to be assigned to each feature.

### C.3   `Python` Implementation

The full end-to-end pipeline of LLM-Lasso is implemented in `Python`. The score collection is done via `OpenAI` APIs for GPT models and o1, and via `OpenRouter` otherwise. `Langchain` is used for the retrieval step of the RAG pipeline.

For computing the data-driven baseline metrics (such as mutual information and MRMR), we first produce a set of randomly generated 50/50 train and test splits and save them to CSV files. These splits are used

for both LLM-Lasso and each data-driven baseline. Then, baseline scores can be computed via our `Python` implementations, relying on `scikit-learn`.

The implementation of the LLM-Lasso model, given the importance scores, is based on the package `adelie` (Yang & Hastie, 2024). We have a custom fork of `adelie` that adds AUROC and misclassification error metrics to the output of `adelie.cv.cv_grpnet`. For the transformation applied to the penalty factors, we consider powers of the inverse importance. Cross-validation, with folds determined by `scikit-learn`'s `StratifiedKFold`, determines which power of the inverse importance to use. Results are averaged across the same folds as used to compute the baselines.

**Output Parsing**   LLMs, especially smaller models can make formatting mistakes. For instance, they may not including all necessary genes, or may include extra genes (e.g., ones mentioned in retrieval context) as part of the genes to score. For `GPT-4o` and `o1`, we use structured outputs to directly receive the scores as a `Python` object. In models where this streamlined score collection feature is not available, we rely on the output formatting from the prompt (see Appendix B) and search for floating point scores that immediately flow the double-asterisk-colon sign (with or without space). If we fail to collect scores, we retry until the correct scores for the batch are collected.

## D   Small-Scale Dataset Details

Table 3: Summary of small-scale experiment datasets.

| Dataset | Year | n | p | Source |
|---|---|---|---|---|
| Spotify* | 2024 | 4600 | 29 | Source |
| Wine | 2009 | 6497 | 11 | Source |
| Diabetes | 1998 | 768 | 8 | Source |
| Bank | 2012 | 45211 | 51 | Source |
| Glioma | 2022 | 839 | 23 | Source |

We source a wide range of small-scale datasets for feature selection in classification and regression. We use an asterisk to indicate the datasets that are released after the cutoff dates for all models sampled (see Table 2 in the main body for an overview of the model cutoff dates). For all small-scale datasets, we remove features whose values are neither numerical nor categorical, and remove rows with missing values (if a column has a significant amount of missing values, we drop that column instead). We remark that the purpose of the small-scale experiment is not meant to demonstrate performance on the specific task but rather to evaluate feature selectors, even in the absence of some potentially informative features and data. As such, we do not, e.g., perform imputation of missing values.

## E   Additional Experimental Results

In this section, we present additional experimental results that did not appear in the main body of the paper.

### E.1   Study of Downstream $\ell_2$ Regularization Heuristic

To further explore the effect of the downstream $\ell_2$-regularized regression heuristic (Section 3.1.2), we perform a brief study on selected small-scale datasets. On a subset of the small-scale datasets, we run Lasso and LLM-Lasso, both with and without the heuristic (using the hyperparameters described in Section 4.1.2). Results on the test set are plotted in Figure 11. We can see that, in most cases presented, (i) without the heuristic, LLM-Lasso outperforms Lasso, (ii) the heuristic improves performance for both LLM-Lasso and Lasso, and (iii) the heuristic can diminish the advantage of LLM-Lasso over Lasso. For instance in Diabetes, Spotify, and Wine without the downstream $\ell_2$ heuristic, the test error curve for LLM-Lasso is generally below that of the plain Lasso. However, when the heuristic is applied, this advantage is reduced for Spotify, and goes away for Diabetes and Wine (for Bank, the AUROC is fairly unchanged). As seen in

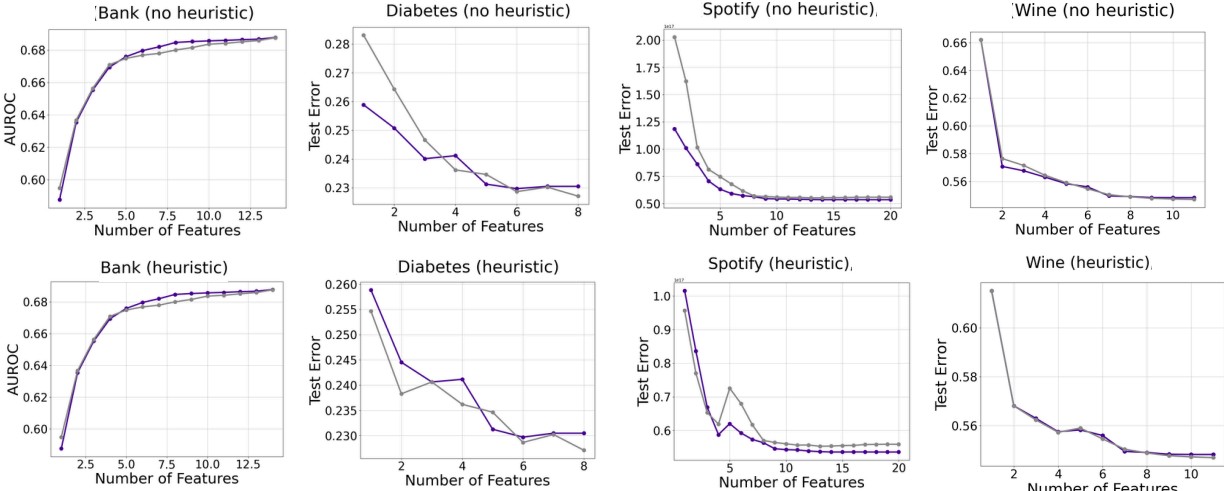

Figure 11: Comparison of Lasso and LLM-Lasso, with and without the downstream $\ell_2$-regularized regression heuristic described in Section 3.1.2. **LLM-Lasso** is plotted in **purple** and **Lasso** is plotted in **gray**.

the large-scale experiments (Figure 5), LLM-Lasso can noticeably outperform Lasso, even with the heuristic, but this appears dataset- or domain-dependent. Possibly, the advantage is overall more apparent for higher-dimensional, under-determined problems like those chosen for the large-scale experiments.

## E.2 AUROC Performance of LLM-Lasso for Large-Scale Experiments

Figure 12 illustrates the AUROC performance of our model against various baseline across the three high-dimensional lymphomal datasets. In most cases, the strong performance demonstrated by Figure 5 carries over to the AUROC metric, though it is not as strong as by the test error metric presented in Figure 5.

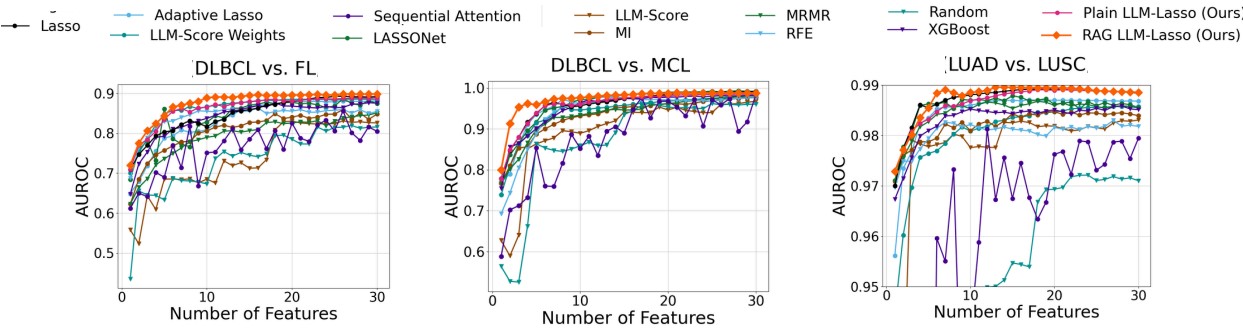

Figure 12: AUROC performance across 10 splits for the lymphoma datasets: LLM-Lasso vs. baselines.

## E.3 Ablation of Model Temperature

We perform ablations over LLM temperature with `temp = {0, 0.5, 1}`, on the MCL vs. DLBCL lymphoma classification task. We find that the test error is agnostic to the temperature for both plain and RAG-augmented LLM-Lasso. Results are plotted in Figures 13a and 13b. The results for plain and RAG-enhanced LLM-Lasso are very similar, with lower test error for RAG.

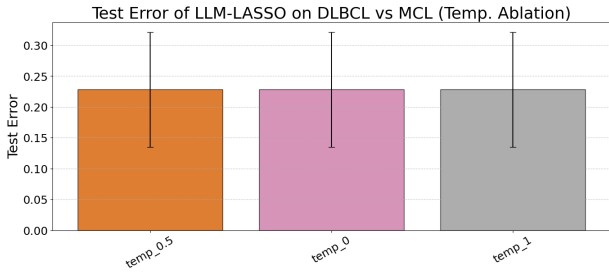 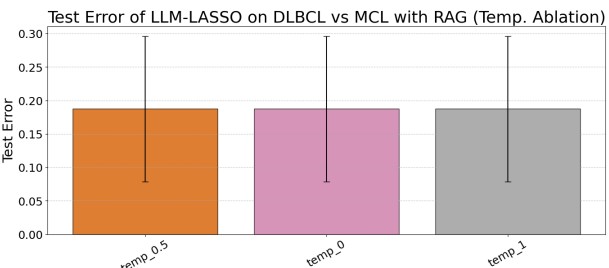

(a) Temperature ablation of Plain LLM-Lasso.   (b) Temperature ablation of LLM-Lasso with RAG.

Figure 13: Temperature ablations of LLM-Lasso, where error bars are standard deviation over 10 random seeds.

### E.4 Recall Study of OMIM RAG System

We evaluate RAG via (i) the recall@k metric, and (ii) impact of RAG on LLM-Lasso accuracy. We evaluate recall for the OMIM vector database described in Section 4.1.2. We first construct 100 retrieval queries by randomly sampling without replacement from the retrieval queries performed during the lymphoma dataset experiments. Then, we compare ground truth nearest documents (by cosine similarity) for each query to those retrieved via Chroma's `query` interface. Results of this experiments are in Table 4. The recall numbers appear reasonable for our purposes; the vector database is consistently retrieving the documents with the closest embeddings, even if not all of the time.

Table 4: Recall@k for the OMIM RAG system

| k | 1 | 3 | 5 | 10 | 25 | 50 |
|---|---|---|---|---|---|---|
| **Recall (%)** | 90.74 | 87.04 | 86.30 | 85.19 | 82.52 | 79.24 |

#### E.4.1 Feature contributions

We perform a study of the top features selected by Lasso, LLM-Lasso (Plain), and LLM-Lasso (RAG) for one of the large-scale biomedical experiments.

In the experiments we run multiple LLM-Lasso regressions, and thus we are unable to extract a single list of selected features and their coefficients. For better interpretability, we introduce a feature contribution metric that takes the proportion that each feature appears across the full path of the number of features. A feature contribution of 1 means the feature appeared in all the models, while that of 0 means the feature appeared in none of the models. We create heatmaps of the union of genes with top 10 feature contributions for the Lasso, Plain LLM-Lasso, and RAG LLM-Lasso, as well as the polarity of the coefficients, represented as letters in the heatmaps ("F" coefficients in the direction of FL and "D" for DLBCL) (Figure 14).

In the clinically relevant problem of classifying FL and DLBCL, there are several genes with high feature contributions that have relevance in cancer genomics and hematology/oncology, especially in the GPT-4o LLM-Lasso heatmap. For example, *AICDA*, *BCL2*, and *BCL6*, all of which have high feature contributions in the RAG LLM-Lasso, have been implicated in the transformation of FL to DLBCL (Lossos et al., 2004; Green et al., 2013). Consistent with our findings, a previous study suggests that high *AICDA* expression is implicated in the generation of mutations in the *BCL2* gene, which was associated with increased risk of the transformation of FL into DLBCL (Correia et al., 2015). In addition, higher expression of *MYC* has been implicated in the transformation of FL to DLBCL (Aukema et al., 2017), which is concordant with the identification of *MYC* as an important feature to classify DLBCL from FL by the LLM-Lasso. On the other hand, plain Lasso does not consistently select such genes.

Interestingly, the o1 LLM-Lasso heatmap, although *AICDA* is included as the top gene in the RAG LLM-Lasso, many of the other genes are less relevant to the DLBCL literature (Pasqualucci & Dalla-Favera, 2018).

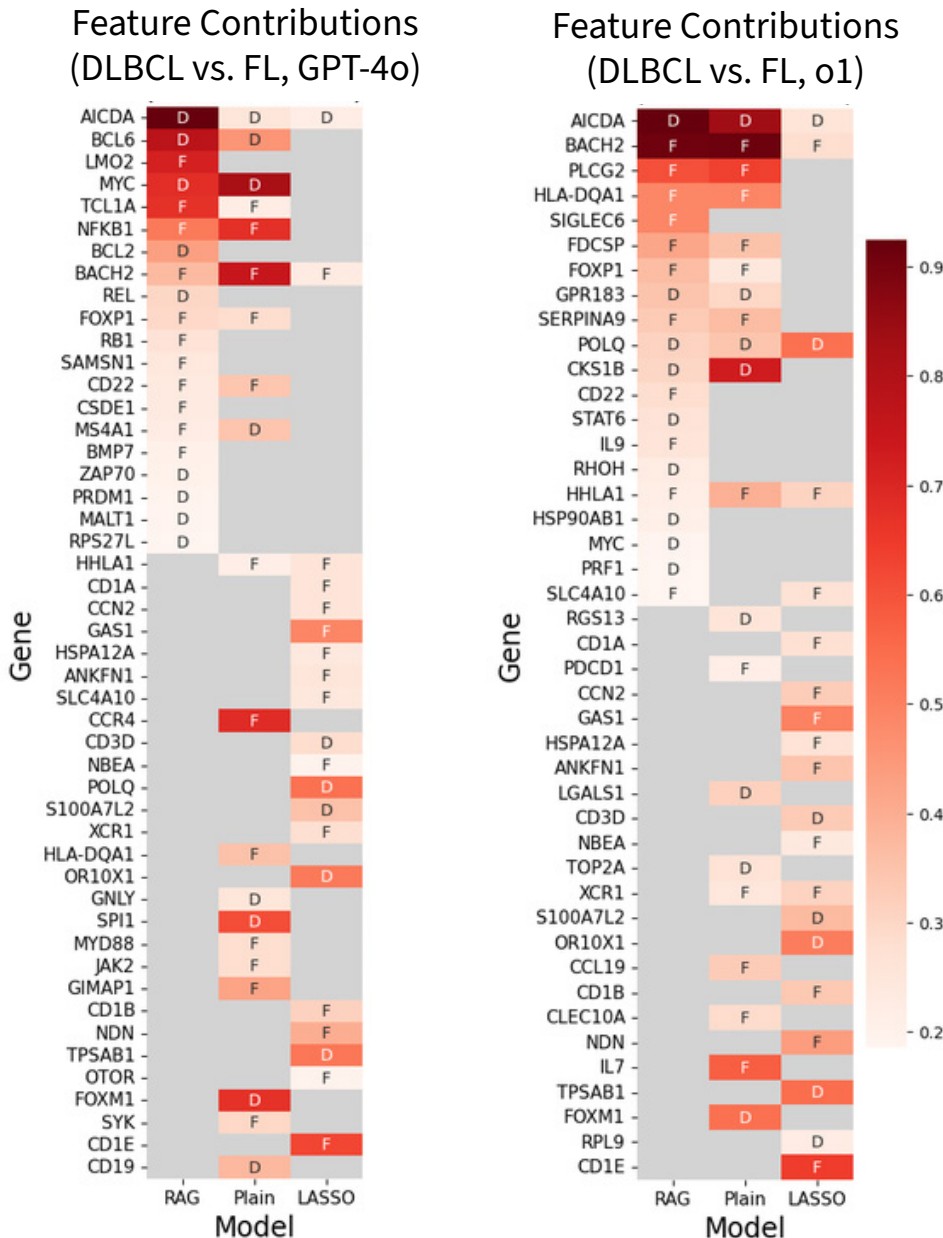

Figure 14: Heatmaps of feature contributions in the FL vs DLBCL experiment for GPT-4o and o1 LLM-Lasso, with and without RAG.

The reason for this contrast with the high accuracy of o1-based LLM-Lasso is unknown; further investigation may be of interest.

### E.5 Example of Hallucinations in Adversarial Experiments

Figure 15 provides an illustrative example of LLM hallucinations in the adversarial experiments, where GPT-4o produces seemingly-plausible explanations for scores assigned to fake genes. This contextualizes the poor performance of LLM-Score in the adversarial experiments of Section 4.4.

**Z8ED**: 0.6
Reasoning: Possible weak connection to cellular proliferation processes affection some cancers.

**PC6LOW**: 0.5
Reasoning: PC6LOW is involved in cell cycle regulation, which can be relevant for distinguishing between DLBCL and MCL.

Figure 15: GPT-4o hallucination for corrupted gene names: an LLM-Lasso penalty factor (left) and an LLM-Score importance score (right). Even though both genes, `Z8ED` and `PC6LOW` are fake, the LLM hallucinates justification for their relevance to the task.

