# OpenReview forum: "LLM-Lasso: A Framework for Domain-Informed Feature Selection and Regularization"
_TMLR — Rejected by TMLR_

### Review · Reviewer_kmwR · 2025-11-23

**Summary Of Contributions:**

The paper introduces LLM-Lasso, an extension of weighted/adaptive Lasso framework in which feature-wise penalties are derived from LLM scores based on feature metadata and task descriptions. A small family of transformations (including a standard Lasso “no-LLM” case) is tuned by cross-validation, so the selected model is never worse than plain Lasso on the validation folds. Extensive experiments indicate improved predictive performance and increased robustness to corrupted feature names compared to baselines.

**Audience:**

Yes

**Audience Explanation:**

The paper sits at the intersection of LLMs, classical regularization, and high-dimensional biomedical modeling, which is clearly relevant for TMLR. The integration of LLM/RAG into Lasso, development of robustness to hallucinations, is both interesting for TMLR’s audience.

**Claims And Evidence:**

Yes

**Claims Explanation:**

The methodology development of this paper is easy to follow, and claims about superior performance and robustness to hallucinations from the LLM are supported by extensive experiments.

That said, I'd suggest authors to tone down the robustness claims a bit. The guarantee that LLM-Lasso “never performs worse than standard Lasso” only actually holds on the cross-validation splits, it doesn’t really extend to unseen test data, especially when you’re working with small sample sizes and lots of features, where cross-validation may overfit. The adversarial corruption experiments are promising but only use one dataset and one way of corrupting feature names.  Overall, the positioning is closer to “we include Lasso in our model family, so in the worst case we can recover Lasso-like behavior” rather than a strong robustness guarantee in the general sense.

**Requested Changes:**

**Q1**. Tone down the robustness claims to make clear that the “never worse than Lasso” property holds on cross-validation splits.

 **Q2**.  I think the method is essentially a special case (extension) of weighted/adaptive Lasso with LLM-generated weights, consider clarifying the relationship to adaptive/weighted Lasso and present LLM-Lasso explicitly as such.

**Q3** Typos : Page 2: “Section 4 determines … and exemplify” should be “exemplifies”. Page 19: “directly discribe” should be “describe”.

---

> ### Author Response · Authors · 2025-12-17
> **Response to Referee kmwR**
>
> Dear Reviewer kmwR,
>
> Thank you for your positive assessment of our manuscript. We greatly appreciate your recognition of the methodology’s clarity, the empirical support for our performance and robustness claims, and the relevance of our work at the intersection of LLMs, classical regularization, and high-dimensional biomedical modeling. We have carefully revised our manuscript following your suggestions.
>
> ---
> ### 1. Tone down the robustness claims to make clear that the “never worse than Lasso” property holds on cross-validation splits.
>
> We agree that the guarantee is specific to cross-validation (CV) data and does not extend to test set generalization, particularly in high-p/low-n regimes where CV can be optimistic. We have toned down the language throughout to emphasize this as a practical safeguard rather than a broad generalization guarantee.
>
> Specifically:
>
> - In the abstract, we now toned down our claim and instead revised: “In the case of poor LLM generation quality, the cross-validation procedure allows LLM-Lasso to recover the behavior of the plain Lasso.”
> - In section 3.1.1, we now qualify our robustness claim explicitly via: “Though this is a basic, straightforward guarantee (and does not ensure generalization to the test set performance), it provides a clear improvement in robustness over other LLM-guided feature selection methods.”
> - In the contributions (at the end of introduction), we reframed: “We encourage robustness by using the data to cross-validate LLM decisions, using k-fold cross-validation to choose from a family of transformations on the LLM-generated penalty factors. We construct the transformation family such that LLM-Lasso can never perform worse than Lasso on the cross-validation data. LLM-based filter feature selectors lack even this basic guarantee. Via adversarial examples in Section 4.4, we show that this can result in Lasso-like performance on the test data if LLM penalties are uninformative.”
>
> ---
> ### 2. Clarify the relationship to adaptive/weighted Lasso and present LLM-Lasso explicitly as such.
>
> Thank you for this insightful point. We agree that framing LLM-Lasso as an extension of weighted/adaptive Lasso better highlights its foundations while underscoring our contributions in LLM integration and CV tuning. We have made this explicit:
>
> In the introduction, we added: “To do so, we take inspiration from Adaptive Lasso (Zou, 2006), which adds feature-specific weights to the $l_1$ regularization term of Lasso regression. Whereas adaptive Lasso commonly selects penalty factors based on weights from linear or ridge regression, LLM-Lasso instead aims to select penalty factors via domain knowledge. Prior works such as (Bergersen et al., 2011; Urda et al., 2018) assign penalties by exploiting external datasets or citation-based weighting to compute penalty factors. We take a similar but potentially more comprehensive and generalizable approach: we leverage large language models (LLM) to generate penalty factors for all predictors using domain-specific insights.”
>
> These changes clarify the lineage without overstating novelty, and we cross-reference the large-scale experiment section (Section 4.2.2) showing improved test accuracy of LLM-Lasso over plain adaptive Lasso in high-dimensional settings.
>
> ---
> ### Q3. Typos.
>
> All noted typos have been corrected (and a full proof-read performed)
>
> ---
> These revisions strengthen the manuscript’s clarity and humility while maintaining its empirical focus. We are grateful for your thorough review and constructive suggestions, which have notably improved the work. We look forward to your further thoughts during discussion.
>
> Sincerely,
>
> The Authors

---

### Review · Reviewer_43yZ · 2025-11-27

**Summary Of Contributions:**

The paper proposes LLM-Lasso, a framework that uses large language models (LLMs) to construct feature-specific penalty factors for $\ell_1$-regularised regression/classification. Instead of treating all coefficients equally as in standard Lasso, the authors ask an LLM (optionally augmented with retrieval over a domain-specific corpus) to assess the relevance of each predictor for a given prediction task, producing a vector of scores $V$. These scores are then mapped to per-feature penalty factors $W = \tau(V)$ via a family of simple transformations (e.g., inverse-importance powers), and a weighted Lasso is fit with penalty $\lambda \sum_j w_j \lvert \beta_j \rvert$. Crucially, the transformation $\tau$ is selected by cross-validation over a small family that includes the trivial “all ones” mapping, so that the procedure can interpolate between “ignore the LLM” (plain Lasso) and “trust the LLM strongly.” This gives a way to use LLM-based prior knowledge in the Lasso penalty. The authors also describe a task-specific prompting and RAG pipeline to obtain LLM scores from feature names and task descriptions, and evaluate LLM-Lasso on several high-dimensional biomedical datasets, where they report improvements over standard Lasso and over previous LLM-based filter-style feature-selection baselines.

**Key strengths (high level):**

- Conceptually simple and practically appealing way to use LLM-derived domain knowledge as a prior over feature importance in a standard Lasso pipeline.
- Experiments on realistic, high-dimensional biomedical tasks show that the approach can yield non-trivial gains over both classical and prior LLM-based baselines.

**Key weaknesses (high level):**

- The methodological novelty is moderate: the main contribution is how to inject LLM-derived scores into an existing weighted Lasso framework, rather than proposing a fundamentally new learning algorithm. It is better viewed as a new learning pipeline built on top of standard tools than as a from-scratch methodological innovation.
- There is limited theoretical analysis of the advantages and limitations of this new learning pipeline.

**Additional Comments:**

Overall, I find the core idea of the paper quite appealing from a practical standpoint: using LLM-derived feature scores to define penalty factors for a weighted Lasso is simple, intuitive, and potentially useful in domains with rich textual prior knowledge. The pipeline is easy to implement and could be of interest to practitioners.

That said, in my view the contribution is too incremental for acceptance at TMLR. The method is best understood as a pragmatic integration of existing components (weighted Lasso, cross-validation, RAG, prompt engineering), rather than a substantively new learning algorithm or a conceptually new framework. On top of that, the paper has serious issues in terms of exposition and notation: key sections are over-framed with non-standard terminology, the mathematical setup is at points imprecise or confusing, and the writing often obscures a relatively simple underlying idea. This gives the impression that the work is not yet mature enough, either conceptually or in terms of presentation, to be taken seriously for publication in its current form. With a clearer, more standard formulation of the learning setup, a more calibrated claims about novelty, and a significant tightening of the text, the work could become a solid applied-methods contribution, but I do not think it meets the bar at this stage.

**Audience:**

Yes

**Audience Explanation:**

Yes. The paper sits at the intersection of several topics that are currently of broad interest to the TMLR audience: how to leverage LLMs as sources of prior knowledge, how to integrate such priors into classical statistical methods (here, weighted Lasso), and how to handle high-dimensional, domain-rich settings such as genomics. Even though the methodological novelty is moderate, many practitioners and researchers working on applied ML, feature selection, and LLM-assisted modelling would likely be interested in the proposed pipeline, the empirical findings, and the practical lessons on prompt/RAG design and cross-validation safeguards.

**Broader Impact Concerns:**

I do not see major ethical risks beyond those typical of LLM-assisted modelling in biomedicine, but a brief Broader Impact section would be appropriate. In particular, the authors should note that (i) the method is not ready for direct clinical use, (ii) LLM-derived priors may reinforce existing biases in the biomedical literature, and (iii) reliance on a proprietary LLM raises transparency and reproducibility concerns.

**Claims And Evidence:**

Yes

**Claims Explanation:**

Overall, the core empirical claims are mostly supported, but some of the higher-level statements are stronger than what the evidence strictly shows. The paper does provide a reasonably clear description of the LLM-Lasso pipeline, compares against sensible baselines (plain Lasso and prior LLM-based feature selection methods), and reports consistent improvements on several high-dimensional biomedical datasets. This supports the claim that LLM-derived penalty factors can sometimes enhance performance and sparsity in realistic scenarios.

However, the evidence is more limited for some of the broader claims. For example, the “never worse than Lasso” guarantee is true by construction for the cross-validation objective, but the paper does not provide a systematic analysis of test-time degradation or failure cases. The choice of transformation families and CV criteria is somewhat ad hoc, and there is little statistical analysis (e.g., uncertainty estimates, robustness checks across seeds or alternative metrics) to quantify how reliable the observed gains are. In addition, the benefits of RAG versus plain prompting, and of different prompt designs, are only partially explored. As a result, while the experimental section is convincing for the main practical message, it is less conclusive for the more general or universal claims about robustness and general advantage.

**Requested Changes:**

I would recommend the following changes and clarifications:

1. **Clarify and correct the framing and notation.** I would strongly encourage simplifying the “supervised data-driven learning” vs “task-specific data-driven learning” terminology and tightening the mathematical notation. At present, these sections feel heavier than necessary for such a standard ERM (Empirical Risk Minimisation) setup, and in places they are not fully precise mathematically. Concretely:
   - The spaces where the data live are not clearly specified (e.g., $\mathcal X$, $\mathcal Y$, and the product space $\mathcal X \times \mathcal Y$), nor is the underlying distribution over these spaces clearly defined.
   - The notation $p(\cdot \mid \theta)$ is introduced but not cleanly connected to the predictive model; the same symbol $\theta$ is used for the data-generating parameter, the model parameter, and the estimator, which blurs the distinction between the true parameter (if any) and the learned parameters.
   - The formalisation of “supervised data-driven learning” in Section 2.1 is, in my view, overly complicated for a standard empirical risk minimisation setup and not entirely clean in terms of notation. The same issue appears in the “task-specific data-driven learning” definition, where a somewhat abstract composition $T(f(D), h(D_{\text{meta}}))$ is introduced without clear benefit and with limited mathematical precision.

   I recommend: (i) rewriting these parts using standard supervised learning notation (ii) clearly separating the roles of “true” parameters, model parameters, and estimators, and (iii) considering whether the labels “supervised data-driven learning” / “task-specific data-driven learning” actually add technical clarity beyond stating that the method is ERM with a metadata-driven regulariser. A simpler and more standard formalisation would make the main contribution (LLM-informed penalty factors for Lasso) much easier to see.

2. **Improve clarity and tone of the exposition.** Overall, the paper is harder to read than it needs to be. The core idea is actually quite simple, using LLM-derived feature scores to define penalty factors for a weighted Lasso, but the presentation is often obscured by fuzzy, high-level terminology and somewhat inflated framing. In several places, broad concepts (“task-specific data-driven learning,” “task-specific LLM,” etc.) are introduced with ambitious wording but only loosely tied to the concrete LLM-Lasso pipeline, which gives the impression of overselling a relatively straightforward idea.

   I would recommend tightening the writing to focus more directly on:
   - the concrete pipeline (prompt/RAG → LLM scores $V$ → transformed penalties $W$ → weighted Lasso + CV),
   - the specific modelling choices (transformation families, CV criterion), and
   - the empirical findings (when and how much LLM guidance helps).

   Reducing vague or grandiose terminology, and replacing it with precise but simple explanations, would make the paper considerably more accessible and would help the main contribution stand out more clearly.

3. **Provide more justification and analysis for the transformation families and CV criterion.** Explain why the chosen families (inverse-importance powers, ReLU-based penalties) are appropriate, and discuss their limitations. If possible, add some sensitivity analysis showing how results change with different families or CV criteria.

4. **Calibrate the claims about novelty and contribution.** The core idea, using LLM-derived feature scores to construct penalty factors for a weighted Lasso, is interesting but conceptually incremental, as it builds on well-established components (weighted Lasso, cross-validation, RAG, prompt engineering). I would suggest making this positioning more explicit: clearly articulate that the main novelty lies in the integration and pipeline design rather than in a fundamentally new learning algorithm or theoretical framework. This could involve (i) softening or clarifying high-level claims that may suggest a stronger form of methodological originality than what is actually delivered, and (ii) sharpening the comparison to prior work on weighted Lasso and LLM-based feature selection so that the specific incremental advances of LLM-Lasso are easier to see.

---

> ### Author Response · Authors · 2025-12-17
> **Response to Referee 43yZ**
>
> Dear Reviewer 43yZ,
>
> Thank you for your thoughtful and constructive review. Your feedback has been invaluable in making the manuscript clearer, more rigorous, and better framed. Below we summarize the major revisions; please see the revised paper for full details.
>
> ---
>
> ### 1. Clarified framing and notation (Section 2.1)
>
> We rewrote Section 2 (“Preliminaries”) using standard supervised-learning notation and simpler language. The prior “task-specific data-driven learning” subsection is now integrated within “Supervised Learning,” avoiding over-framing. We explicitly define the subspaces of $ \mathcal{X} $ and $ \mathcal{Y} $, remove all discussion of underlying distributions and $p(\cdot|\theta)$, and use $\theta$ only for model parameters. We eliminated unclear references to the true data-generating parameter and instead describe the direct mapping from data to predictions. Excess notation has been reduced, including in the LLM sections, though cross-validation notation remains (we welcome further suggestions if needed).
>
> ---
>
> ### 2. Improved clarity and tone
>
> We tightened the exposition substantially—removing phrases like “transformative opportunity” and “seamlessly integrates,” and using precise, factual language. The core idea is now presented concisely in one introductory paragraph (alongside the pipeline figure). We consolidated methodological details previously scattered across sections into a single unified section (plus some further configuration details in the beginning of the experimental section), moving only long components (e.g., prompts, computational details) to the appendix.
>
> ---
>
> ### 3. Added justification and analysis for transformation families and CV criterion
>
> We introduced a new Subsection 3.2.1 (“Transformation Family Simulation Results”) showing results for $\eta \in \{0,\dots,10\}$ and adding a ranking-based family (the ReLU transformation) that sets penalties of the top $(1-\gamma)p$ features to 0 and scales the rest linearly by rank. We hypothesize why inverse-importance factors may outperform the ReLU family and note that further exploration of penalty families is warranted.
> We also include an ablation (Appendix E.1) studying the downstream $\ell_2$ regularization heuristic and its interaction with cross-validation.
>
> ---
>
> ### 4. Calibrated novelty and contribution
>
> We now frame LLM-Lasso as a *practically useful pipeline* built on established components, notably the Adaptive Lasso. The introduction now reads:
>
> > “We take inspiration from Adaptive Lasso (Zou, 2006)… whereas Adaptive Lasso derives weights from linear or ridge regression, LLM-Lasso instead uses domain knowledge via LLMs. Prior works (Bergersen et al., 2011; Urda et al., 2018) similarly use external data or citation-based weighting; we extend this by leveraging LLMs for domain-informed penalty factors.”
>
> All robustness claims are now explicitly limited to cross-validation robustness, using moderated phrasing such as “encourages robustness” or “can be more robust.”
>
> ---
>
> ### Broader Impact
>
> We added a short statement noting that:
> - The method is not validated for clinical use;
> - It may amplify biases present in biomedical corpora;
> - Reproducibility may be limited due to reliance on proprietary LLM APIs.
>
> ---
>
> We believe these revisions fully address your concerns and greatly strengthen the paper.
> Thank you again for your time and thoughtful feedback.
>
> Sincerely,
> The Authors

---

### Review · Reviewer_AsnE · 2025-11-30

**Summary Of Contributions:**

# Summary

This work introduces LLM-Lasso, a new method for incorporating LLM-derived domain knowledge into feature selection. This works by considering a weighted version of the Lasso regularization penalty, determining weights via LLM elicitation and a cross-validation procedure. The resulting procedure is tested in low-dimensional and high-dimensional settings, where LLM-Lasso consistently performs well compared to baselines, including several ablated variants.

# Strengths
- It's timely and interesting to consider LLM-augmented versions of classical statistical models such as Lasso. Applications to biomedical data, where classical feature selection techniques typically struggle, has a particularly high potential for impact, and is well-suited to LLM capabilities.
- LLM-Lasso is principled, and follows quickly from other weighted Lasso variants. The cross-validation procedure to essentially determine LLM trustworthiness is a nice touch.
- The results show impressive performance with respect to sensible baselines. In particular, LLM-Lasso with RAG consistently outperforms adaptive Lasso and other LLM-augmented feature selection schemes.

# Weaknesses
- I think the writing can be improved. Much of the exposition is a bit verbose, and a lot of details are left to the appendices, at times referencing multiple different appendices in the same sentence. Additionally, the ordering of Sections 4 and 5 is odd to me. I think it would improve the manuscript substantially to be more succinct and include more details in the main body of the text.
- Restriction to the linear model setting is a limitation; adapting LLM-Lasso techniques to more general model classes is another place where meaningful improvement seems possible.
- Much of the experimental evidence is gene expression data; understanding the performance of LLM-Lasso in other scenarios could be helpful to practitioners.

**Audience:**

Yes

**Audience Explanation:**

There is clear audience for LLM-augmented statistical models. The submission in particular makes a strong case that practitioners in biomedical sciences may find the results appealing.

**Claims And Evidence:**

No

**Claims Explanation:**

I find the claims generally well-supported by the presented evidence, with notable exception to the claim that "consequently, irrespective of the LLM’s generation quality, LLM-Lasso is guaranteed never to perform worse than standard Lasso." This is not what is guaranteed by cross-validation (unless performance is exclusively measured on the training set), and the claim should be revised accordingly. I would be comfortable changing my answer to "yes" with this set of claims toned down.

**Requested Changes:**

- **Critical:** Revise claims about guaranteed superior performance.
- **Critical:** Revisit some of the writing, and in particular, the ordering of sections 4 and 5.
- Consider applications to additional fields.
- As a minor comment, at the beginning of page 5, `\citet` should be used for the in-text citation.

---

> ### Author Response · Authors · 2025-12-17
> **Response to Referee AsnE**
>
> Dear Reviewer AsnE,
>
> Thank you for your thoughtful and encouraging review.
> We address your main comments below, with corresponding references to the revised manuscript.
>
> ---
>
> ### 1. Claims about guaranteed superior performance
>
> We have systematically qualified and clarified this claim throughout the paper. For example:
>
> - **Abstract:**
>   > “In the case of poor LLM generation quality, the cross-validation procedure allows LLM-Lasso to recover the behavior of the plain Lasso.”
>
> - **Section 3.1.2:**
>   > “Though this is a basic, straightforward guarantee (and does not ensure generalization to test data), it provides a clear improvement in robustness over other LLM-guided feature selection methods.”
>
> - **Contributions:**
>   > “We construct the transformation family such that LLM-Lasso can never perform worse than Lasso itself on the cross-validation data. LLM-based filter feature selectors lack even this basic guarantee…”
>
> These revisions make it explicit that the guarantee is *cross-validation–based only*.
> We also adjusted the experimental discussion to emphasize both strengths (e.g., high-dimensional settings) and weaknesses (e.g., small-scale datasets).
>
> ---
>
> ### 2. Writing, succinctness, and organization of Sections 4 and 5
>
> We substantially edited and streamlined the manuscript:
>
> - Removed verbose phrasing and redundant explanations (see also response to Reviewer 43yZ).
> - Simplified mathematical notation and replaced unclear formulations with plain English where appropriate.
> - Consolidated and reordered material for better flow:
>   - Combined Sections 4 and 5 into a single **“Experiments”** section.
>   - Moved simulation results and heuristic descriptions from the appendix to the main text.
>   - Organized experiments by importance: large-scale → small-scale → adversarial → ablations.
> - Restructured the appendix for readability, removing unused or tangential content.
>   Overall length reduced from **30 to 26 pages.**
>
> ---
>
> ### 3. Applications to additional fields
>
> We agree that broader applications would strengthen the work. Our focus remains on *high-dimensional, low-sample* domains where LLM-derived priors are especially valuable—biomedical data serve as a representative case.
> A small non-biomedical experiment is included to show generalizability, though the main gains appear where domain knowledge is crucial.
> We now clarify this rationale in the **Introduction** and **Discussion**, and the new **Section 6 (Broader Impact)** highlights the need for broader future evaluations.
>
> ---
>
> ### 4. Restriction to linear models
>
> We agree this is a limitation and now explicitly note it as a direction for future work.
> We retain the linear setting for clarity, interpretability, and ease of analysis of LLM-derived penalties, but acknowledge that extending LLM-Lasso to nonlinear models is a natural next step.
>
> ---
>
> ### 5. Minor comment: citation format
>
> All citations now use `\citep` consistently. Thank you for catching this.
>
> ---
>
> We believe these revisions address your comments and significantly improve both clarity and scope.
> Thank you again for your generous and constructive feedback.
>
> Sincerely,
> The Authors

---

### Review · Reviewer_TpdK · 2025-12-01

**Summary Of Contributions:**

The paper proposes LLM-Lasso, a variant of Lasso where each feature receives its own penalty coefficient, derived from scores produced by an LLM using feature names and task description. A small family of transformations maps these scores to penalty multipliers; cross-validation selects both the transformation and the regularization strength, and the family includes the standard unweighted Lasso as a special case. The method is tested on low-dimensional benchmarks and high-dimensional gene expression datasets, and the authors also study the effect of retrieval augmentation and adversarial corruption of feature names.

Strengths:
Clear mechanism to inject LLM-based prior information into a standard regularized model; extensive empirical study, including an experiment with corrupted feature names; good performance on challenging biomedical datasets.

Weaknesses:
No theoretical analysis; robustness claims are stated more strongly than justified; the scope of experiments is mostly gene expression; dependence on prompt and RAG design is only partially explored.

**Audience:**

Yes

**Audience Explanation:**

The work lies at a point where several active areas intersect: the use of LLMs as sources of prior information, feature selection in high dimensions, and applications to genomics and related biomedical problems. The proposed construction is easy to integrate into existing Lasso workflows and demonstrates that textual knowledge can be used in a controlled way via cross-validation. This is likely to interest readers working on applied ML with LLMs, regularization, and domain-informed models.

**Broader Impact Concerns:**

I do not see major ethical concerns.

**Claims And Evidence:**

Yes

**Claims Explanation:**

The main empirical claims are supported: the proposed method is described in sufficient detail, compared to sensible baselines, and the reported results show consistent gains on the main biomedical tasks as well as reasonable behavior under adversarial feature-name corruption. The relationship to existing weighted/adaptive Lasso methods is straightforward and correctly handled. However, the wording that suggests LLM-Lasso is “guaranteed not to perform worse than Lasso” goes beyond what is proven: the guarantee holds for the cross-validation criterion on the available folds, not for out-of-sample performance in general. If that statement is softened and made precise, the evidence matches the claims.

**Requested Changes:**

Robustness statement.
Revise all statements about “never worse than Lasso” to explicitly refer to the cross-validation objective (on the validation folds) and not to unseen test data. A short clarification in the main text would be sufficient.

Experimental scope and sensitivity.
If space permits, either (a) add at least one non-genomic dataset with rich feature names, or (b) provide a small sensitivity study on one dataset (e.g., different RAG settings or prompt variants) to illustrate robustness of the design choices.

---

> ### Author Response · Authors · 2025-12-17
> **Response to Referee TpdK**
>
> Dear Reviewer TpdK,
>
> Thank you for your careful reading and for the positive overall assessment. We address your two requested changes below.
>
> ---
> ### 1. Robustness statement---“never worse than Lasso”
>
> We have revised every occurrence of this claim to make it explicit that the guarantee holds only for the cross-validation objective on the validation folds and does not extend to unseen test data. Key changes include:
>
> - **Abstract:**
>   > “In the case of poor LLM generation quality, the cross-validation procedure allows LLM-Lasso to recover the behavior of the plain Lasso.”
>
> - **Section 3.1.2:**
>
>   > “Though this is a basic, straightforward guarantee (and does not ensure generalization to test set performance), it provides a clear improvement in robustness over other LLM-guided feature selection methods.”
>
> - **Contributions:**
>
>   > “We construct the transformation family such that LLM-Lasso can never perform worse than Lasso itself on the cross-validation data. LLM-based filter feature selectors lack even this basic guarantee. Via adversarial examples in Section 4.4, we show that this can result in Lasso-like performance on the test data if LLM penalties are uninformative.”
>
> These revisions (also noted in our responses to the other reviewers) fully address the overstatement.
>
> ---
> ### 2. Experimental scope and sensitivity
>
> We agree that broader empirical evidence would be valuable. However, the strength of LLM-Lasso depends critically on the informativeness of the LLM-generated scores, which in turn requires (i) meaningful textual feature names and (ii) accessible domain knowledge (either encoded in the LLM or supplied via RAG). Constructing a domain-specific RAG pipeline and curating new datasets with rich, interpretable feature names is unfortunately beyond the scope of a revision cycle.
>
> Instead, we strengthened the manuscript by clarifying and foregrounding existing evidence that already speaks to generality and robustness:
>
> 1. Generalizability beyond gene expression.
>
> Section 4.3 includes small-scale experiments (~20 features) on UCI datasets with interpretable textual feature names. LLM-Lasso performs competitively with plain Lasso and prior LLM-based baselines in these non-genomic settings, demonstrating that the method extends to domains where feature semantics are meaningful, though it is not always the highest-performing model and its advantage is limited in the small-scale case.
>
> 2. Sensitivity to design choices.
>
> We agree that additional ablation studies and sensitivity analyses would further strengthen the paper. Given resource constraints and the short revision period, we were unable to conduct a full sensitivity analysis of the prompting strategy or RAG system design. However, we have expanded our analysis where feasible. On the Lasso side, we now highlight the existing comparison of penalty-transformation families (previously in the appendix but now moved into the main paper), and we include new analysis examining an important heuristic: applying a downstream $\ell_2$-regularized regression when the selected feature set is small.
>
> For the RAG component, Appendix E (“Additional Experimental Results”) includes a recall study of our OMIM-based retrieval system across multiple top-k configurations for the largest biomedical task, as well as an ablation of model temperature. While a comprehensive study of sensitivity to prompting and retrieval design is beyond the scope of the current revision, we explicitly acknowledge this limitation in the conclusion and identify it as an important direction for future research.
>
> We also clarify the motivation for focusing on biomedical examples in the introduction and discussion. Specifically, our focus is high-dimensional, low-sample settings where domain knowledge is both abundant and especially valuable, and we chose the biomedical domain as a representative setting where these conditions hold and where supervised learning problems are practically important.
>
> ---
> Thank you again for your constructive and insightful feedback, which has noticeably strengthened the work.
>
> Sincerely,
>
> The Authors

---

### Review · Reviewer_fmK3 · 2025-12-01

**Summary Of Contributions:**

This work proposes a method to inject Large Language Model (LLM) based regularization into a Machine Learning (ML) training pipeline. More specifically they propose a framework to guide feature selection for Lasso ($l_{1}$) Regression by using domain specific information extracted from LLMs which is then converted to an appropriate weight penalty to be used in composition with Lasso Regression. This is further improved with an optional Retrieval-Augmented Generation (RAG) pipeline which is used to provide additional information as part of prompt preparation for this process. This along with features present in the data are then fed to the LLM to generate penalty scores which are then used in an iterative Cross Validation process adapting those scores into lasso penalties to learn the best model over the validation set.\
The process of adaption of scores is essentially a **Guess-and-Check** approach which transforms the LLM scores using a family of pre-set transformations $\tau$, like ***inverse importance family***. The reason for using such transformations appears to help in creating a search space in which to look for better penalties on lasso weights, as the pipeline of the experiment doesn't include the LLM(s) in any (Closed loop) feedback architecture (Though the system prompt can be considered as a limited form of feedback). That is, once an LLM generates the score, its role is exhausted and the learning process the revolves around iterating over each member of the transformation set $\tau$ to discover the best performing penalties on the cross validation loss. Including an multiplicative identity transformation ($\eta = 0$) provides a way for the pipeline to select the baseline model - Lasso Regression, in case the search fails.\
The main contribution of this work is to provide a simple way to integrate LLM based feature selection information into parameter learning for the regression model. Through ablations the authors try to demonstrate the advantage of LLM based scoring for improved performance and Cross Validation based feature transformation selection for improved robustness.

**Audience:**

Yes

**Audience Explanation:**

The paper offers a useful platform for utilizing LLMs to incorporate background knowledge in prediction tasks and many ablations in the paper demonstrate the utility of doing so, especially when aided with Retrieval Augmented Generation. This work is in line with many other attempts in the area of tabular or time series forecasting to incorporate additional background information to make the models perform better or be robust. The platform is simple to implement as detailed in the appendix and can provide a useful platform to build and improve upon.

**Broader Impact Concerns:**

N/A.

**Claims And Evidence:**

No

**Claims Explanation:**

The utility of incorporating additional information LLMs in the problem setting stems from a combination of improving performance or model understanding. The proposed framework uses LLMs to essentially perform a structured sparsity inducing regularization where the structure is imposed based on the knowledge extracted from an LLM. As reference baselines in addition to Lasso, wouldn't other sparsity inducing penalties [1,2] help establish an advantage of the proposed approach? Given the sometimes relative marginal performance advantage (and sometimes deficit) as shown in Fig. 3b or Fig 4, wouldn't it be more suitable to also compare a data-dependent LLM induced sparsity with another (pre-set/ non-data dependent) sparsity prior in addition to Lasso?.\
As mentioned in the paper, the goal of feature selection is to improve model performance and the proposed methodology tries to learn the appropriate transformation function so that LLM-Lasso performance can be bounded by Lasso in the cross-validation loss, however in many of the results like for instance in adversarial simulations and small scale studies and some large scale datasets, See Fig 3,4,14,15, the performance of Lasso is sometimes better than LLM-Lasso, and in some cases LLM Lasso defaults to using Lasso. Given such a finding what would be a necessarily meaningful utility of LLM Lasso over Lasso for any user of this framework?\
When comparing large scale studies, one clear advantage of LLM Lasso is that unlike LMPriors, it is not prohibitive to use in a high dimensional setting, which definitely makes the method meaningful. However, when comparing LLM-Lasso to say a method like XGBoost, wouldn't a fairer baseline also be works like LLM-Boost/PFN-Boost[3] which aim to improve the performance of Decision Trees (GBDT) using LLM based scores to train the underlying DT.\
Optionally, although not directly comparable but still relevant, could the authors compare their method with or comment about Tabular Prior Fitted Networks[4]. As TabPFNs use incontext-learning to make predictions and don't need additional training, but if they indeed perform better on the unpublished Lymphoma Data and the Lung Cancer data, that would be useful context for a user of LLM-Lasso to bear in mind.




### References

[1] Francis Bach, Rodolphe Jenatton, Julien Mairal, & Guillaume Obozinski. (2011). Optimization with Sparsity-Inducing Penalties.\
[2] Junzhou Huang, Tong Zhang, & Dimitris Metaxas. (2009). Learning with Structured Sparsity.\
[3] Mayuka Jayawardhana, Renbo, Samuel Dooley, Valeriia Cherepanova, Andrew Gordon Wilson, Frank Hutter, Colin White, Tom Goldstein, & Micah Goldblum. (2025). Transformers Boost the Performance of Decision Trees on Tabular Data across Sample Sizes.\
[4] Han-Jia Ye, Si-Yang Liu, & Wei-Lun Chao. (2025). A Closer Look at TabPFN v2: Understanding Its Strengths and Extending Its Capabilities.

**Requested Changes:**

If the authors could answer or run experiments (based on space and compute constraints) about baselines and claims made in the paper, that would improve the manuscript. Additionally if the authors could judiciously use the 12 pages allocated in the main body to either make the paper more self contained without repeat readings needed from the appendix, like Section 5) or the figures more legible to parse results properly (Like Fig 5) it would make the paper more understandable and hopefully reduce difficulty in reading for readers at large.

---

> ### Author Response · Authors · 2025-12-17
> **Response to Referee fmK3**
>
> Dear Reviewer fmK3,
>
> Thank you for your detailed and thoughtful review. We address your main concerns below.
>
> ---
>
> ### 1. TabPFN-v2, PFN-Boost, and LLM-Boost as baselines
>
> We appreciate the reviewer bringing up these strong tabular methods. We agree that using LLMs directly for tabular prediction (TabPFN-v2) or as tabular models within boosting frameworks (PFN-Boost, LLM-Boost) can be powerful in supervised learning.
>
> Our work instead focuses on *augmenting classical statistical learning with external semantic knowledge* absent from the dataset. This design has a major computational advantage: tabular LLM methods must process each datapoint individually, whereas LLM-Lasso uses the LLM only once to generate feature relevance scores, after which adaptive Lasso handles all training and prediction.
>
> LLM-Lasso differs conceptually as well: PFN-Boost and LLM-Boost act as black-box predictors mapping $X$ to $Y$ directly, while LLM-Lasso retains a convex, interpretable estimator whose penalties are modulated by LLM-derived signals. As a result, PFN-Boost and LLM-Boost do not directly apply to feature selection (the LLM directly predicts $Y$), whereas we explicitly focus on a feature selection in our work. This separation makes it possible to isolate and study how semantic priors interact with data-driven learning and feature selection, something not feasible with end-to-end LLM tabular models.
>
> We have added discussion of TabPFN-v2, PFN-Boost, and LLM-Boost to the introduction. Because they operate under a different modeling paradigm, we do not include them as baselines in our feature-selection-focused evaluation.
>
> ---
>
> ### 2. Sparsity-inducing penalties and structured-sparsity baselines
>
> The reviewer’s cited works on sparsity-inducing penalties and structured sparsity are highly relevant. The first (“Optimization with Sparsity-Inducing Penalties”) outlines a broad family of objectives, including Lasso, regularized by such norms. While we focus here on the widely used $l_1$ penalty, *LLM-generated feature weights could in principle extend to other sparsity-inducing norms*, and we now mention this as a future direction.
>
> The second (“Learning with Structured Sparsity”) covers methods like group Lasso that rely on predefined feature structures. Our current setting lacks known groups or hierarchies; instead, LLM-Lasso infers feature-specific relevance directly from metadata. Nonetheless, the paradigm naturally extends to structured sparsity (e.g., using LLMs to propose groups or graph constraints), which we now note in the conclusion.
>
> For this paper, we therefore focus on unstructured $l_1$ sparsity and compare against a diverse set of baselines (classical, LLM-based, filter, wrapper, and embedded) aligned with our feature-selection objective.
>
> ---
>
> ### 3. Utility of LLM-Lasso when it defaults to plain Lasso
>
> This behavior is *intentional and advantageous*. LLM-Lasso is designed as a *safe wrapper* around Lasso:
>
> - When LLM scores are informative (e.g., large-scale biomedical tasks), performance improves substantially (Figure 5).
> - When scores are noisy or misleading, cross-validation automatically selects the null transformation, reverting to plain Lasso. In adversarial corruption experiments (Figure 8), where feature names are randomized, LLM-Lasso maintains reasonable performance, unlike baselines such as LLM-Score, which can collapse.
>
> This conditional adaptivity allows practitioners to deploy the method safely even when uncertain about LLM quality in new domains. We emphasize this strength more clearly in the revised Introduction and Discussion.
>
> For small-scale tasks where domain knowledge adds little beyond the data, LLM-Lasso’s improvement is modest; we now make it clear in the revision that LLM-Lasso is not expected to be the best method across all possible domains. In particular, it is best suited to high-dimensional domains (e.g., biomedical) rather than purely data-rich ones.
>
> ---
>
> ### 4. Readability and presentation
>
> We improved the manuscript’s clarity and presentation by:
> - enhancing figure legibility (especially Fig. 5),
> - moving key appendix material into the main text, and
> - shortening and restructuring Section 5.
>
> See also our response to Reviewer AsnE for additional detail.
>
> ---
>
> Thank you again for your careful reading and constructive feedback. Your comments have noticeably strengthened the final manuscript.
>
> Sincerely,
> The Authors

---

### Author Response · Authors · 2025-12-09
**Request for Extension**

Dear Dr. Rawat,

To ensure we can devote sufficient time to a detailed and thorough revision of the paper, we would like to request a one-week extension for submitting our revised manuscript and responses to the reviewers (until December 21st). Thank you for your consideration and for serving as the Action Editor for our paper.

Best regards,

Authors of LLM-Lasso

---

### Author Response · Authors · 2025-12-17
**A Note on the Revision**

Dear Reviewers and AC,

We have re-run the Lasso component of some experiments to ensure that the hyperparameters and CV metric were consistent with what was stated in the body of the paper. As such, there may be minor differences in the plots between the original submission and revised manuscript. This is due to modified Lasso hyperparameters, not any fundamental changes.


Sincerely,

*The Authors*

---

### Decision · Action_Editor_bYb9 · 2026-01-20

**Recommendation:** Reject

**Additional Comments:**

The reviewers raised three main points in their initial reviews:

1) Revising some of the main claims to better reflect the empirical evidence provided and better contextualizing the contributions of the submission;
2) Expanding the scope of the experiments by considering high-dimensional datasets from additional domains; and
3) Providing a comprehensive treatment of prompt and RAG component design

While the authors adequately addressed the first point, a considerable amount of additional work is required to address the remaining two points. The authors are encouraged to resubmit their work after resolving these remaining weaknesses highlighted by the reviewers.

**Audience:**

Yes

**Audience Explanation:**

As all the reviewers agree, the paper studies a timely problem of leveraging LLMs for a classical problem of feature selection, which the TMLR community should find interesting. That said, a more comprehensive empirical evaluation of the proposed method would strengthen the contributions and attract a wider audience.

**Claims And Evidence:**

No

**Claims Explanation:**

The paper proposes an LLM-based approach for feature selection in Lasso $\ell\_1$ regression. The utility of the proposed approach, namely LLM-Lasso, is validated on high-dimensional datasets from the biomedical domain. LLM-Lasso, when combined with a RAG-based component, outperforms various baselines. The authors also show the competitiveness of LLM-Lasso on small-scale datasets. Furthermore, LLM-Lasso is shown to be robust compared to alternative LLM-driven methods in a specific adversarial setting.

Since incorporating RAG into LLM-Lasso is crucial for obtaining improved performance in the high-dimensional settings considered in the paper, a more comprehensive treatment of this topic is essential to strengthen the contributions of the submission. Additionally, expanding the scope of the empirical study beyond the biomedical domain in a high-dimensional setting would better engage the interest of the broader TMLR community.

**Resubmission Of Major Revision:**

The authors may consider submitting a major revision at a later time.